# Single-Atom Co-Catalysts Employed in Titanium Dioxide Photocatalysis

**Ujjaval Kerketta [1,†]** , **Alexander B. Tesler [1,†]** **and Patrik Schmuki [1,2,3,*]**

1    Department of Materials Science and Engineering, Institute for Surface Science and Corrosion WW4-LKO, Friedrich-Alexander-Universität Erlangen-Nürnberg, 91058 Erlangen, Germany
2    Chemistry Department, King Abdulaziz University, Jeddah 80203, Saudi Arabia
3    Regional Centre of Advanced Technologies and Materials, Palacky University, Listopadu 50A, 772 07 Olomouc, Czech Republic
*    Correspondence: schmuki@ww.uni-erlangen.de
†    These authors contributed equally to this work.

**Abstract:** With a distinct electronic structure and unsaturated coordination centers, supported single-atoms (SAs) have shown great potential in heterogeneous catalysis due to their superior activity, stability, and selectivity. Over the last few years, the fascination of SA-use spread also over photocatalysis, i.e., a particular case of heterogeneous catalysis in which chemical reactions are activated by charge transfer from an illuminated semiconductor. Titanium dioxide ($TiO_2$) is one of the most studied photocatalytic materials. It is widely used as a light absorbing semiconductor decorated with metallic (nanoparticles and single-atom) co-catalysts. In the current review, we emphasize the role of SAs as a co-catalyst in photocatalysis, and clearly set it apart from the use of single atoms in classic heterogeneous catalysis. The review first briefly describes the principal features of SAs, and gives an overview of most important examples of single-atom co-catalysts. Then, we discuss photocatalysis and key examples of single-atom co-catalysts used on $TiO_2$ photocatalysts and their applications. At last, we provide an outlook for further exploring $TiO_2$-based single-atom photocatalytic systems.

**Keywords:** single-atom co-catalyst; photocatalysis; titanium dioxide; $H_2$ evolution; water splitting; $N_2$ fixation

## 1. Introduction

Catalytic processes are not only at the heart of many industrial processes, such as the production of fuels as well as bulk and fine chemicals in all branches of industry, but also can be used for the decomposition of pollution and energy generation [1]. The main reason to use catalysts is that they offer alternative and energetically advantageous reaction mechanisms to allow processes to be carried out at industrially favorable ranges of pressure and temperature. Catalysts may be employed to stimulate reactions in a variety of surroundings from gases to liquids and solid surfaces. The catalytic reaction typically occurs on a molecular/atomic level on small catalytically active particles, or reactive sites, where the rupture of bonds in reactants and the formation of new bonds occur.

In contrast to thermally activated catalysis, in photocatalysis, a particular case of heterogeneous catalysis, the reaction is triggered by photoexcited charge carriers. Here, the catalytic activity is provided by a light absorber that allows the creation of electron-hole ($e^-$-$h^+$) pairs, followed by their separation and transport to the surface for reaction. In other words, the typical heart of a photocatalyst is a semiconductor. Photocatalytic characteristics were already discovered at the beginning of the 20th century on ZnO [2]. However, the real breakthrough in photocatalysis research occurred in 1972, when Fujishima and Honda demonstrated electrochemical photolysis of water on a $TiO_2$ electrode irradiated by UV light [3]. Photocatalytic $H_2$ production is still, and more than ever, the main research direction [4].

H$_2$ is perceived as one of the feasible solutions to an efficient energy source and storage, based on a green and sustainable energy vector [5]. Although the concept of direct sunlight to H$_2$ conversion is most attractive, the performance of traditional photocatalytic systems is still unsatisfactory, mainly due to complexities related to photocatalytic materials, i.e., serious electron-hole pair recombination in the semiconductor, and sluggish charge transfer on the semiconductor surface. Hence, to improve the overall catalytic H$_2$ production efficiency of commonly used photocatalytic systems, the semiconductor is usually decorated by a suitable co-catalyst(s) [6]. Co-catalysts that stimulate H$_2$ evolution reaction (HER) are typically noble metal nanoparticles (NPs) (Pt, Pd, Rh, etc.) [7]. However, the application of noble metals as co-catalysts is hampered due to low natural abundance resulting in their high costs. Furthermore, their utilization efficiency in the form of NPs is low since only surface atoms of the NP participate in the catalytic reaction [8]. There are several directions to reduce noble metal consumption: (i) replacement of expensive noble metals by e.g., MoS$_2$ or TiN [9], (ii) design of core-shell bimetallic (PtCu) [10,11], or tri-metallic (AuAgCu) co-catalyst structures [12,13], and/or (iii) minimize costly metal cluster size down to single metal atoms. The latter can maximize metal atom efficiency while still maintaining necessary catalytic performance [14,15].

Using single-atom co-catalysts (SACs) has the potential not only to provide a high utilization of noble metal atoms (ideally 100%), but may also introduce novel and high selectivity due to unusual reaction schemes supported at SA sites [16]. The SA issue is dating back to 2003, when Flytzani-Stephanopoulos and coworkers reported cationic single-atom Au species supported CeO$_2$ for the water-gas shift reaction [17]. The authors demonstrated comparable or even higher activity of single-atom Au-supported CeO$_2$ compared to a fully decorated Au NPs CeO$_2$ catalyst. In 2011, Qiao et al. [18] deposited single Pt atoms on iron oxide (FeO$_x$) for CO oxidation. Experimental and theoretical studies demonstrated a high catalytic activity of Pt SA that was attributed to the partially vacant 5$d$ orbitals of Pt, while positively charged, high valent Pt atoms reduced the CO adsorption energy as well as the activation barriers for CO oxidation. This is in contrast to bulk Pt surfaces known to be readily poisoned by CO due to its strong adsorption.

The first reports on SAs used in photocatalysis date back to 2014. Xing et al. [19] reported single-atom (Pt, Pd, Rh, and Ru) anchoring on anatase TiO$_2$ for photocatalytic H$_2$ evolution reaction. The SA co-catalyst (SACs) exhibited higher catalytic activity toward H$_2$ evolution as compared to the NP counterparts. Theoretical studies revealed higher activity due to favorable H* adsorption/desorption at the Pt-O sites [20]. Later works were focused on C$_3$N$_4$ support to anchor SA as a model to understand its role in photocatalytic reactions [21–23]. So far, a range of SACs (Ir, Pt, Au, Rh, and Pd) have been prepared on various substrates, such as TiN, Al$_2$O$_3$, TiO$_2$, ZnO, Co$_3$O$_4$, FeO$_x$, CeO$_2$, CdS, ZnIn$_2$S$_4$, zeolite, and semiconductive metal-organic frameworks (MOFs), showing apparent catalytic activity toward diverse reactions [24–33]. In 2016, Liu et al. [34] decorated ultrathin TiO$_2$(B) nanosheets with Pd SA (Pd$_1$) by photodeposition approach. The optimized Pd$_1$/TiO$_2$ catalyst displayed high activity and selectivity toward the hydrogenation of C=C bonds. Since then, various SACs have been synthesized and tested for photocatalytic applications such as H$_2$ evolution [14,35,36], CO$_2$ reduction [37–41], N$_2$ fixation [42–45], organic synthesis [46], and degradation of pollutants [47–49].

In the present review, we will focus our discussion on SA co-catalysts deposited on a TiO$_2$ substrate to form an effective photocatalyst system. In photocatalysis, TiO$_2$ is the classic benchmark semiconductor material that due to band edge positions is suitable for a wide range of different reactions and applications. In photocatalytic redox reactions, e.g., the conduction band electrons can reduce water or nitrogen compounds to form H$_2$, NH$_3$, or O$_2{}^\bullet$, while the valence band holes can oxidize, e.g., water or nitrogen compounds, forming OH$^\bullet$, O$_2$, or NO$_3$. Among available photocatalytic materials, TiO$_2$ is abundant, non-toxic, and has outstanding photocorrosion stability. Therefore, we will first point out the differences between a co-catalyst approach in photocatalysis as compared to conventional catalysis by discussing the basic principle of photocatalysis and indicating the main

drawbacks associated with the materials commonly used for photocatalytic purposes. Then, we will introduce the concept of co-catalysts as a feasible solution to overcome some of the intrinsic complexities associated with semiconducting photocatalytic materials. Next, single-atom co-catalysts will be compared to NPs and their stability on $TiO_2$ support will be discussed. We will further demonstrate the application of various SACs deposited on $TiO_2$ nanostructured supports to be exploited for photocatalytic applications. Finally, challenges facing single-atom co-catalyst design, as well as their different perspectives, will be discussed.

## 2. The Fundamental Principle of Photocatalysis

There are several substantial differences between classical catalysis and photocatalysis. In a chemical catalytic process, the catalytic material is ready to react. However, in a photocatalytic process, the photocatalyst needs to be externally stimulated, i.e., by the ability of the photocatalyst to absorb solar radiation (e.g., co-catalytic reaction rate may be fully determined by photo-generated charge carriers' availability) [50]. Another substantial difference is that in chemistry, thermodynamics dictates the reaction direction meaning that in the case of negative Gibbs free energy change, $\Delta G < 0$, the reaction would proceed spontaneously, while utilization of catalyst decreases the activation energy of the intermediate states increasing the reaction rate [51]. In the case of photocatalysis, however, the absorption of light supplies required external energy/stimuli to drive even thermodynamically uphill reactions, such as photocatalytic water splitting [52].

The key component of photocatalytic systems is the semiconductor. The activity of a photocatalyst depends on several factors as schematically shown in Figure 1a. (1) *Light-harvesting*: the bandgap determines the light absorption wavelength range of semiconductor material. When light is irradiated on a photocatalyst with energy equal to or greater than the bandgap, an electron is excited from the valence (VB) to the conduction (CB) band and electron-hole ($e^-$-$h^+$) pairs are generated. (2) *Separation of charge carriers*: the charge carriers are separated (under the field of Schottky junction at the semiconductor-electrolyte interface, or by carrier diffusion. (3) *Transfer*: the generated charge carriers reach the semiconductor-electrolyte interface to react with suitable redox levels in the solution [53]. An ideal semiconductor for a specific photocatalytic reaction should simultaneously fulfill several of the aforementioned tasks starting from efficient light absorption followed by charge carrier separation and transfer and, finally, should allow for fast kinetics of the desired surface reaction (e.g., $H_2$ or $O_2$ evolution). In a most ideal case, the semiconducting material should be abundant, low-cost, as well as resistant to photocorrosion. So far, no semiconducting material, even for the water-splitting case, has been found to match the satisfaction of all these demanding characteristics. Therefore, the development of composite (combination of different materials) photocatalysts or photoelectrodes has been used to address different points [53].

The capability of a semiconductor to carry out photoinduced electron-hole transfer to surface-near redox species is governed by the band energy positions of the semiconductor relative to the redox potential of the adsorbate. The energy level at the bottom of the conduction band (CB edge) represents the electron reduction potential, while that at the top of the valence band (VB edge) determines the hole oxidizing capability. From a thermodynamic point of view, adsorbed species can be reduced (photocatalytically) by CB electrons if they have more positive redox potentials than CB edge, and, correspondingly, can be oxidized by VB holes if they have more negative redox potentials than the VB edge. From a kinetic point of view, for a semiconductor photocatalyst to be effective, the rates of the different interfacial processes involving electrons and holes must be higher than those of the recombination processes in the bulk or at the surface [50]. For instance, thermodynamically, in the water-splitting reactions, the reduction potential of water should be lower than the CB, and the oxidation potential should be higher than the VB of the semiconductor. In view of kinetics, the rate of these charge transfer reactions is very important, as $e^-$$h^+$ pairs undergo recombination in the semiconductor bulk, or via surface

states [54]. Charge recombination can be radiative (photon emission) or non-radiative (heat emission) processes but, in any case, leads to severe losses in the photon conversion efficiency [55]. Charge carrier recombination is enhanced if the chemical bonding of the periodic crystal lattice is altered, i.e., at crystal defects or the surface of the material. The weaker bonding at such sites reduces splitting between bonding and antibonding orbitals compared to valence and conduction band states and thus gives rise to electronic states within the semiconductor band gap [56]. Such intraband gap states can influence charge carrier dynamics offering pathways for the energetic relaxation of photogenerated charge carriers in the valence or conduction band. These states are therefore generally referred to as "trap states", or as "surface states" if they occur on surface sites [54]. In $TiO_2$, particularly in anatase, surface recombination occurs mainly via an intrinsic surface state, while such intrinsic defects could alter and/or create additional energy levels that may have a positive effect on photocatalytic processes and, therefore, are of great importance [57]. However, their exact role in photocatalysis is controversial. From one side, surface states have been claimed to facilitate charge separation, localizing charges close to the material interface, potentially enhancing reactivity. On the other side, they enhance recombination losses being detrimental to overall photocatalytic performance [58,59]. Here, desired product formation (e.g., $H_2$) and recombination processes compete kinetically, i.e., the faster carriers can react off from the semiconductor, the fewer charge carriers recombine. In other words, efficient electron transport/transfer/reaction reduces recombination losses improving the overall conversion efficiency. Particularly the charge carrier transfer/reaction sequence can be substantially accelerated by the use of co-catalysts (as discussed below in more detail) [60].

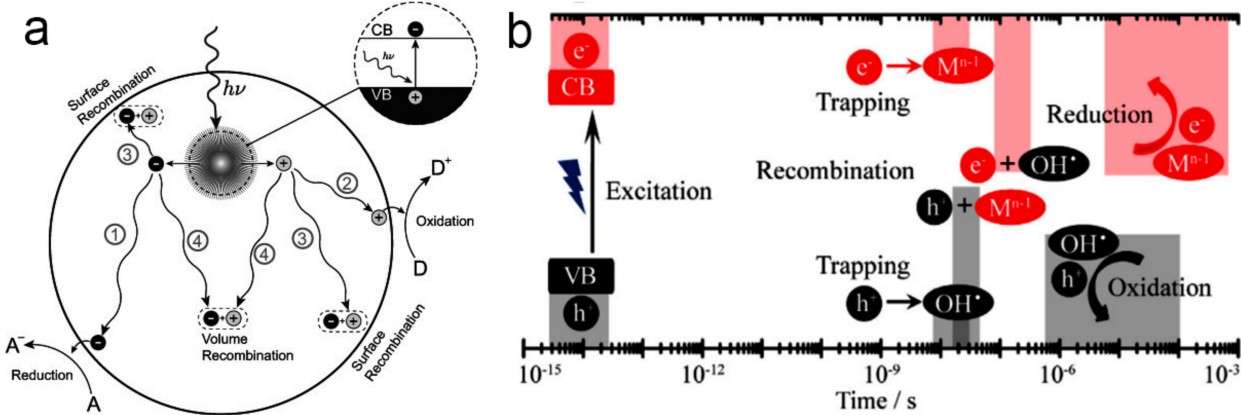

**Figure 1.** (**a**) A general principle of photocatalysis and elementary steps involved in photocatalytic reactions starting from light absorbance and generation of electron-hole pairs to some of the de-excitation pathways for the electrons and holes. Adapted with permission from Ref. [61]. (**b**) A typical time scale for elementary steps in photocatalysis. Adapted with permission from Ref. [62].

In view of the semiconductor, not only the material, but also its geometry from 0D to 1, 2, and 3D architectures has a tremendous influence on the overall performance of photocatalytic systems. The latter not only may change the light-harvesting mechanism by introducing new absorption/reflection pathways, but also shorten the charge separation and transport to the semiconducting interface and enlarge the overall available surface area for photocatalytic redox reactions. For more details, the reader is referred to selected other literature [63–65].

Considering the $H_2$ generation reaction, a wide range of semiconductor materials, such as $\alpha$-$Fe_2O_3$ [66], $TiO_2$ [67], $WO_3$ [68], $BiVO_4$ [69], CdS [70], and $ZnIn_2S_4$ [71], have been investigated. Still, one of the main investigated systems remains $TiO_2$ owing to its photocorrosion stability, appropriate CB and VB energy levels, non-toxic nature, and relatively large abundance [72]. One of the most important benefits of $TiO_2$ is its superior photocorrosion

resistance since the majority of the investigated semiconducting materials have only limited resistance to photoinduced corrosion. The bandgap of anatase $TiO_2$ of ~3.2 eV is the main drawback of the material, as it can only absorb in the UV spectral range of sunlight. The activity of the $TiO_2$ photocatalyst depends on the time scales of the involved photoinduced processes and the catalytic reactions at the surface of the photocatalyst (Figure 1b). The excitation of $e^-$-$h^+$ pairs is fast, i.e., of the order of femtoseconds [62] compared to the subsequent processes that are much slower. For instance, the recombination of electrons occurs in a time interval of 10–100 ns in rutile [62,73,74] and up to a few ms in anatase [75], and that of holes in a time interval of 10–100 ns in both crystalline phases [75,76]. At the same time, the interfacial charge transfer occurs within the time interval of $10^{-6}$–$10^{-3}$ s [62]. These processes are similar to or slower than the recombination of $e^-$-$h^+$ pairs at the surface (anatase) and interior of the bulk material (rutile), while time-resolved spectroscopic measurements reveal that majority of the photogenerated $e^-$-$h^+$ pairs (~90%) recombine after excitation [76]. The latter explicitly demonstrates the intrinsic low photocatalytic activity of $TiO_2$ (and other common semiconducting materials) resulting in quantum yields being considerably below 10%, since charge carrier recombination dominates the entire process [67].

The overall photocatalytic efficiency is determined not only by the competition between charge carrier recombination and trapping (pico- to nanoseconds) but also by the subsequent competition between the recombination of the trapped charge carriers and interfacial charge transfer (micro- to milliseconds) [77]. Such photogenerated charge carriers can be trapped in bulk or migrate to the surface and then be trapped there for the subsequent redox reaction [78]. These recombination/trapping processes depend strongly not just on the origin of the semiconductor, but also on its polymorphic form. For instance, $TiO_2$ exists in three crystal phases, i.e., brookite, anatase, and rutile, among which anatase and rutile are the most stable polymorphs. Anatase is normally considered the most photoactive form due to the longer lifetimes of photogenerated charge carriers [79,80], while rutile displays higher recombination rates [54]. Wang et al. [81] measured trapping processes for anatase and rutile by photoluminescence under weak excitation. Different charge trapping characteristics were obtained for both polymorphs [81]. While anatase shows a visible emission (radiative) due to the donor-acceptor recombination (shallow traps on the surface), where oxygen vacancies and hydroxyl groups serve as the donor and acceptor sites, respectively, rutile displayed strong NIR luminescence (non-radiative) due to the recombination of trapped electrons with free holes, while the trapped electrons are formed by direct trapping or trap-to-trap hopping (deep traps in the bulk) (Figure 2) [78]. The long lifetime of free electrons in anatase was found to be beneficial for reduction reactions, while deeply trapped electrons in rutile elongate the lifetime of holes stimulating multiple hole processes such as water oxidation. Yet, such deeply trapped electrons in rutile do not participate in reduction reactions and do not contribute to the overall activity [82].

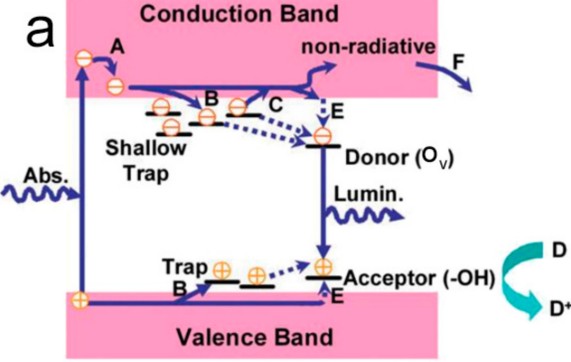

**Figure 2.** *Cont.*

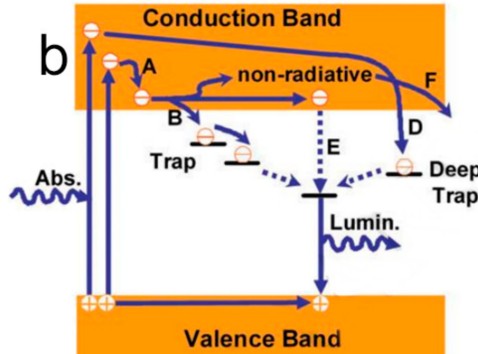

**Figure 2.** Trapping and recombination of photogenerated charge carriers of (**a**) anatase and (**b**) rutile $TiO_2$. Adapted with permission from Ref. [78].

## 3. Accelerating Charge Transfer and Reaction Selectivity of Photocatalyst

### 3.1. Co-Catalyst Approach in Photocatalysis

A most effective way to overcome the sluggish reactivity of common semiconducting materials is by the creation of an electric heterojunction (a suitable Schottky junction), that aids fast surface electron trapping, e.g., electron transfer onto Pt surface is much faster than recombination. Such electron trapping Schottky junctions are formed when a semiconductor is in direct physical contact with metals of a sufficiently high work function. Noble (Pt, Pd, Rh, Au, Ag) and transition metals (Cu, Ni, Co) have been utilized as effective electron trapping co-catalysts; these are conventionally used in the form of nanoparticles decorated on the $TiO_2$ surface [83]. In general, co-catalysts play two key roles in the enhancement of the charge kinetics in such hybrid photocatalytic structures: (i) to trap charge carriers promoting better $e^-$-$h^+$ separation, and (ii) to serve as an active reaction site to supply the trapped charges for redox reactions on their surface (this e.g., is particularly the case for Pt, Rh, Pd, which are classic electrocatalysts for the $H_2$ evolution reaction (HER)) [84]. Generally, the catalysis of $H_2$ evolution reaction (HER) is described according to the Sabatier principle, i.e., the binding strength of adsorbates on the catalyst surface determines the rate of reaction [85]. For the HER, the adsorption of $H^+$ vs. the release of $H_2$ are key factors represented in volcano plots. In particular, Pt is the top catalyst to convert $H_2O$ to $H_2$ based on an optimum of the adsorption of the reactive hydrogen intermediates ($H^*$) and desorption of the products after the reaction.

The Gibbs free energy of hydrogen adsorption ($\Delta G_{H^*}$) is a descriptor of the HER activity of a co-catalyst. For an ideal co-catalyst, $\Delta G_{H^*}$ must be thermoneutral, i.e., equal to zero. Here, $|\Delta G_{H^*}| = 0$ corresponds to the best catalyst for $H_2$ production [86], while the smaller the $|\Delta G_{H^*}|$, the better the $H_2$ evolution activity of a catalyst [87]. When the exchange current density versus $\Delta G_{H^*}$ is plotted, a volcano plot is obtained validating the Sabatier principle (Figure 3a) [88]. However, it was demonstrated that the SA can further optimize the $\Delta G_{H^*}$ and then improve the HER activity. Hu et al. [89] calculated the hydrogen adsorption free energy on $TiO_2$ nanosheets with and without Pt co-catalyst. In the photocatalytic reduction reaction without Pt, $H^*$ tends to adsorb at $O_{2C}$ sites in plain (s-) and defective (Def-s-) $TiO_2$ nanosheets [89]. At this site, however, the $\Delta G_{H^*}$ value is too negative and $H^*$ is strongly adsorbed. On the other side, $\Delta G_{H^*}$ of Pt SA/Def-s-$TiO_2$ is significantly closer to the optimal value of zero ($\Delta G_{H^*} = 0.036$ eV) as compared to Pt NP/s-$TiO_2$ ($\Delta G_{H^*} = -0.402$ eV (Figure 3b). This implies optimal $H^*$ adsorption conditions, indicating that the $H_2$ generation on the Pt SA site is easier than that on the Pt NP.

It was also demonstrated that dual-origin SAs can further tune the electronic properties for the optimal adsorption of the adsorbent. Wang et al. [90] designed Co and Pt dual-SAs with oxygen-coordinated Co-O-Pt dimer sites (13.4% dimer sites). The $\Delta G_{H^*}$ for isolated Co and Pt sites showed a more negative value compared to the benchmark value of Pt (111) in contrast to Co-O-Pt. Dimer sites showed less negative $\Delta G_{H^*}$ value indicating weakened binding to $H^*$ (Figure 3c). The strong binding $H^*$ of isolated Pt and Co can be explained

by high partial density of states (PDOS) edges and localized electronic states, while the Co-O-Pt dimers showed delocalized electronic states resulting in weakened binding toward H*, increasing HER activity.

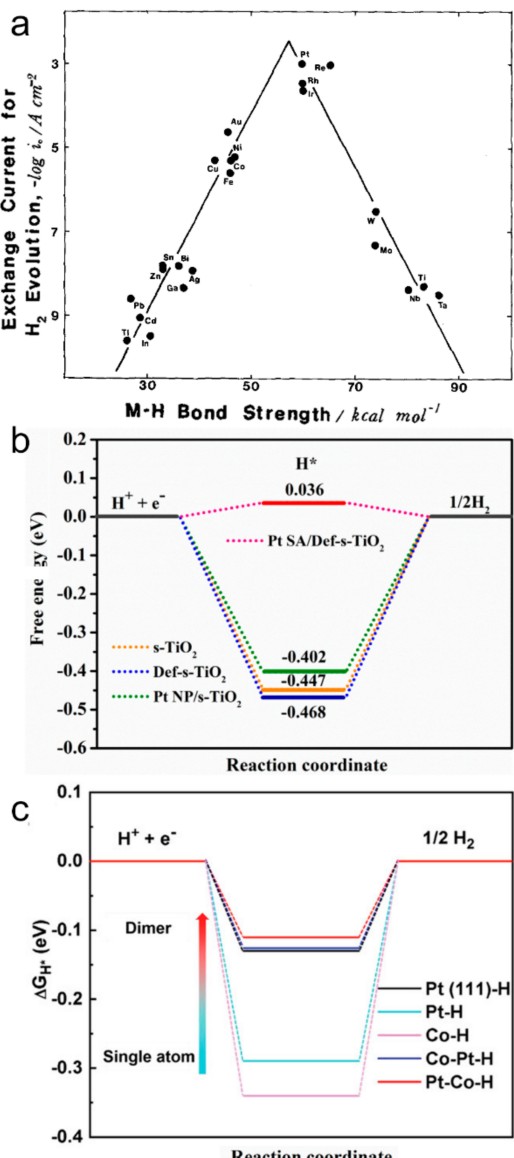

**Figure 3.** (**a**) Exchange currents for electrolytic hydrogen evolution vs. strength of intermediate metal-hydrogen bond formed during electrochemical reaction itself. Adapted with permission from Ref. [84]. (**b**) DFT calculations of free energy $\Delta G_{H^*}$ for H adsorption on s-TiO$_2$, Def-s-TiO$_2$, Pt SA/Def-s-TiO$_2$, and Pt NP/s-TiO$_2$ surface. Adapted with permission from Ref. [89]. (**c**) Gibbs free energy of H adsorption on Co single-atom, Pt single-atom, Co–O–Pt dimer, and bulk Pt (111) sites. Adapted with permission from Ref. [90].

### 3.2. Absence of Schottky Junction in SA Co-Catalysts

According to the classic considerations of a semiconductor-metal junction, the Fermi level at the junction site is lowered to the metal Fermi level due to the much higher charge density in the metals, and depletion of charge carrier takes place in the semiconductor—a Schottky junction is formed [91]. However, for small nanoclusters and SAs, this assumption is not valid since not enough metal states are available to form the Schottky junction [92]. As a result, for SACs, a Schottky barrier as such does not exist. The difference between the SA and small nanoparticles was shown by Niu et al. [42]. Ru SAs were anchored on defect-rich

TiO$_2$ nanotubes (Ru-SAs/Def-TNs) and Ru NPs supported catalyst (Ru-NPs/Def-TNs). Using ultrafast transient absorption and photoluminescence spectroscopy, they studied the relationship between catalytic activity and photoexcited electron dynamics. In PL spectra, Def-TNs showed a spectral profile with a dominant emission at ~420 nm accompanied by a relatively weak emission at ~500 nm, assigned to the band-edge and defect-state emissions, respectively (Figure 4a). The introduction of Ru-NPs lowers the PL intensity to a certain extent, while PL quenching was nearly complete for the Ru-SAs case (Figure 4b). This illustrates that the attachment of Ru in a junction, or even more as a SA, reduces substantially the interband radiative recombination of photoexcited carriers.

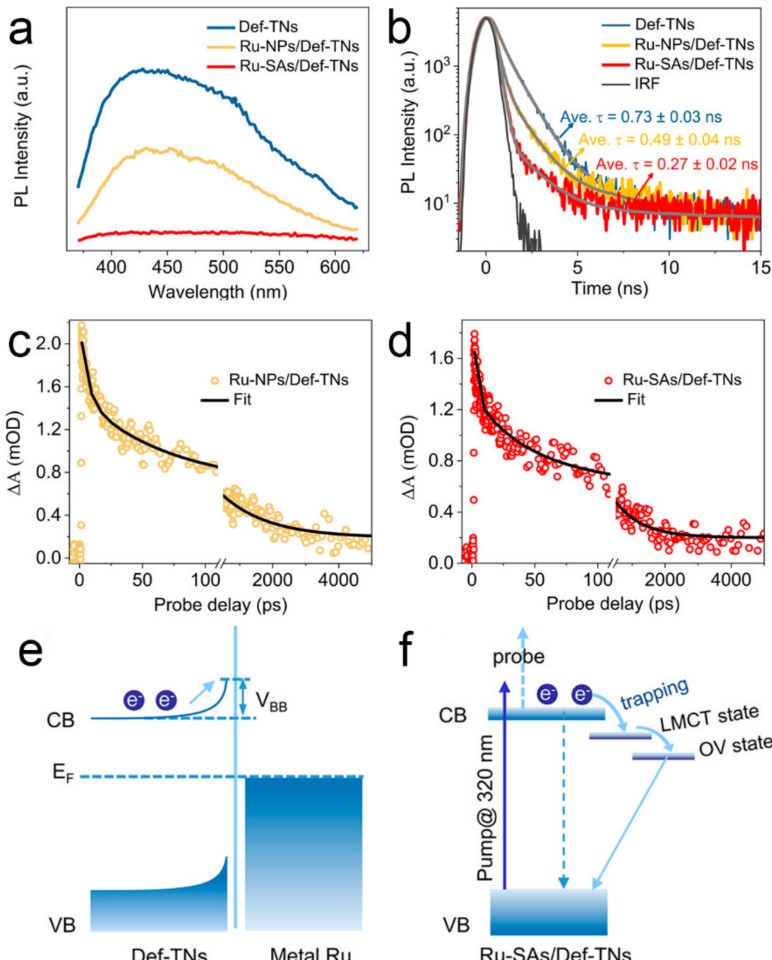

**Figure 4.** (**a**) PL emission spectra and (**b**) time-resolved PL kinetics (excitation at 320 nm, emission at 420 nm) for Def-TNs, Ru-NPs/Def-TNs, and Ru-SAs/Def-TNs. The solid gray lines are the triexponential fits after deconvolution with the instrument response function (black). (**c**) Ru-NPs/Def-TNs and (**d**) Ru-SAs/Def-TNs (probing at 630 nm). Mechanisms underlying the photoexcited electron dynamics involved in (**e**) Ru-NPs/Def-TNs and (**f**) Ru-SAs/Def-TNs, where VB, CB, E$_F$, V$_{BB}$, LMCT, and OV denote the valence band, conduction band, Fermi level, Schottky barrier, ligand-to-metal charge transfer, and oxygen vacancy, respectively. Adapted with permission from Ref. [42].

Ultrafast transient absorption (TA) spectroscopy demonstrated the relaxation dynamics of photoexcited carriers. The average relaxation lifetime of Ru-SAs and Ru-NPs on the Def-TNs was 2.7 and 1.5 times larger, respectively, than that of the bare Def-NTs, indicating a more efficient electron transfer from Def-TNs to Ru-SAs. The electron-trapping capability of Ru-SAs and Ru-NPs on the Def-TNs exhibited a 3.2- and 1.6-fold increase relative to Def-TNs, respectively. The relaxation kinetics of photoexcited electrons in Ru-NPs on the Def-TNs showed a typical feature of electron transfer from semiconductor to metal

in a semiconductor-metal heterojunction (Figure 4e) [42], caused by the formation of a Schottky junction between the Ru metal and Def-TNs with a space charge region at the interface as a result of the electron-hole diffusion [93]. However, in the case of the Def-TNs decorated by Ru-SAs, the coordination of Ru-SAs with the O atoms of the support forms a covalent structure in Ru-SAs/Def-TNs. The electron transfer was promoted indicating electron-trapping assigned to the ligand (Def-TNs) to metal ($Ru^{\Delta+}$) charge-transfer state (LMCT) in the Ru-SAs/Def-TN samples, located in between the CB of Def-TNs and above the $O_V$-related state, which acts as an intermediate trap state (Figure 4f).

*3.3. SAs and Reaction Selectivity*

Size reduction of metallic co-catalyst has been shown to benefit its performance in several aspects [94]. (i) Low-coordination environment of metal centers that is attributed to the unsaturated metal atoms in small metal clusters [95]. (ii) Quantum size effects, i.e., discrete energy level distributions due to confinement of electrons [96]. (iii) Metal-support interactions, which originate from the chemical bonding effect between metal and support lead to a better charge transfer between a metal co-catalyst and support [97]. In SAs, the central metal atoms are stabilized by coordination bonds with e.g., O, N, or S atoms within the support matrix [98]. The electronic and geometric structures of central metal atoms can be then adjusted by tailoring the coordination environment that, in turn, could change the adsorption energy of reactants on metal atoms and thus influence the catalytic pathways. For example, Qiao et al. [18] demonstrated that enhanced activity of $Pt_1/FeOx$ originated from the improved activation of $O_2$ because of the promotion of oxygen-vacancy formation, and the poisoning by CO adsorption was reduced due to the $Pt^{\Delta+}$ active center on the support. As a result, all the elementary steps in the catalytic cycle were exothermic, with low enough barriers for CO oxidation. Besides, the modified electronic properties of metal single active sites were also suggested to be the origin of the high activity of SACs in some other catalytic processes [94].

**4. Synthesis and Anchoring of SAs**

Several techniques have been developed during last decade to form SAs on various photocatalysts. In this review, we limit the synthesis of SACs in view of $TiO_2$ as a photocatalyst, while for general methods on the synthesis of SAs, we refer the reader to the published literature [99–107]. Generally, the metal precursors undergo adsorption, reduction, and confinement by the vacancy defects of the support material [108]. However, single atoms have high surface energy and, therefore, tend to aggregate during the synthesis and further catalysis processes. The latter means that the amount of SAs needs to be optimized since aggregation potentially leads to co-catalyst deactivation [109]. Still, the link between the atomic scale structure of active sites and their reactivity has been comparatively underdeveloped due to the difficulty in characterization techniques to conclusively differentiate atomically dispersed metal species and small metal clusters in various states [59]. The main attempts have been made to somehow maximize the loading of atomically dispersed species to increase the overall quantum efficiency, while high loadings possibly result in atomically dispersed metal species occupying a distribution of coordination environments to the oxide support. Furthermore, SAs can adopt a range of local coordination environments, which dynamically form in response to varied environmental conditions showing a strong influence on their chemical reactivity and catalytic performance limiting the development of detailed structure-property relationships [58]. Such a discrepancy between SAC activity and loading was demonstrated by Qin et al. [110] using a cyanide leaching process of Pt NPs and SAs on $TiO_2$ support. Neither Pt NPs nor the majority of Pt SAs contribute significantly to the co-catalytic activity of Pt on $TiO_2$. In fact, more than 90% of Pt from a standard deposition, i.e., nanoparticles and even small clusters, can be etched away without photocatalytic hydrogen evolution activity loss, indicating that the local coordination environment may play a crucial role in the photocatalytic activity of SACs.

To form the SAC layer, traditional high vacuum vapor deposition techniques have been adopted to deposit SAs, such as size-selected cluster deposition [111] or, in some cases, atomic layer deposition (ALD) [112,113]. Size-selected cluster deposition techniques include an ion source in gas form and a mass spectrometer to filter the nanoclusters according to mass, resulting in metal clusters with a precise number of atoms to be deposited [114]. ALD is a vapor-phase deposition technique based on the subsequent (pulsing) process of precursor molecules. Here, a precursor is chemically adsorbed on the support surface by gas-solid reactions forming an atomic monolayer. By optimizing the ALD conditions, SACs can be designed for a wide range of catalytic materials [115]. The main bottleneck of high vacuum deposition techniques, however, is their relatively high cost of equipment and operation, as well as limited scalability. Therefore, wet chemical approaches are more suitable to be used routinely to synthesize SAs. Upon the system design, wet chemical synthesis of SAs can be achieved by either a bottom-up or top-down approach. In the former, SAs are anchored to the substrate by the reaction of the metal precursor with the anchoring sites on the substrate surfaces. Here, highly dilute metal precursors ($>10^{-6}$ M) are used as a source of the SA, while the substrates are immersed in it to be decorated by SAs. Impregnation, galvanic replacement, electrostatic adsorption, and photochemical adsorption are among the most common bottom-up methods for SA fabrication [114]. In the top-down approach, metallic NPs deposited on the support surface undergo dispersion to form SAs [108]. To obtain SAs, high-temperature atomic migration is typically applied based on Ostwald ripening approach [116]. Under the application of high temperature, metal atoms disconnect from NPs diffusing on the substrate and finally either merge into bigger particles or anchor to defect sites of a photocatalyst [117–119]. Such a high-temperature atomic diffusion technique provides thermally stable SAs, while its applicability and scalability are questionable due to very high temperature and the requirement for an inert atmosphere.

As we discussed above, the stability of SAs under working conditions is one of the most critical concerns in SA catalysis. The atomic diffusion, aggregation, and disintegration during the catalyst preparation usually occur via Ostwald ripening, leading to the growth of large NPs that finally reach thermodynamic equilibrium. Therefore, strong bonding is essential to obtain long-term photocatalytic stability. There are several types of anchoring sites used to form SAs: (i) defects on the support material, (ii) unsaturated coordinated atoms, and (iii) excess oxygen atoms on the support surface to form hollow sites [114].

The chemical potential thermodynamic model was used to explain theoretically the stability of SAs [120]. To form stable SAs, the chemical potential of metal NPs should be greater than that of SAs, $\mu_{NP} > \mu_{SA}$, while to inhibit nucleation of metal clusters, the chemical potential of NPs should be smaller than that of double-atom (DA) clusters, $\mu_{NP} < \mu_{DA}$ (Figure 5a). Han et al. [121] investigated the stability of various noble metals (Ag, Au, Pd, and Pt) on TiO$_2$ using a chemical potential-based thermodynamic model. Figure 5b,c compares the calculated chemical potentials of NPs and SAs on the (001) and (101) TiO$_2$ facets. At higher loading densities on the (001), the chemical potential of SAs is higher than that of NPs, indicating an inability to form SAs. However, $\mu_{SA}$ decreases rapidly with the reduction of loading density enhancing metal-TiO$_2$ binding strength, and then, the metal SAs are more stable for Ag, Au, and Pd. However, at the two loading densities on the (101), $\mu_{SA}$ was much greater than $\mu_{NP}$, and no downward trend in $\mu_{SA}$ was observed, indicating the inability to form the SA, which is consistent with the findings reported by Xing et al. [19]. The latter indicates that even though the (101) facet might stabilize the SAs at a very low loading density, it will be impractical since such a low loading results in the formation of a negligible amount of active sites. The stability of the SA catalyst on the clean (001) facet follows the order Ag ≈ Pd > Pt > Au. It should be noted, however, that Au SACs on the (001) tend to coalesce, leading to the growth of small Au clusters.

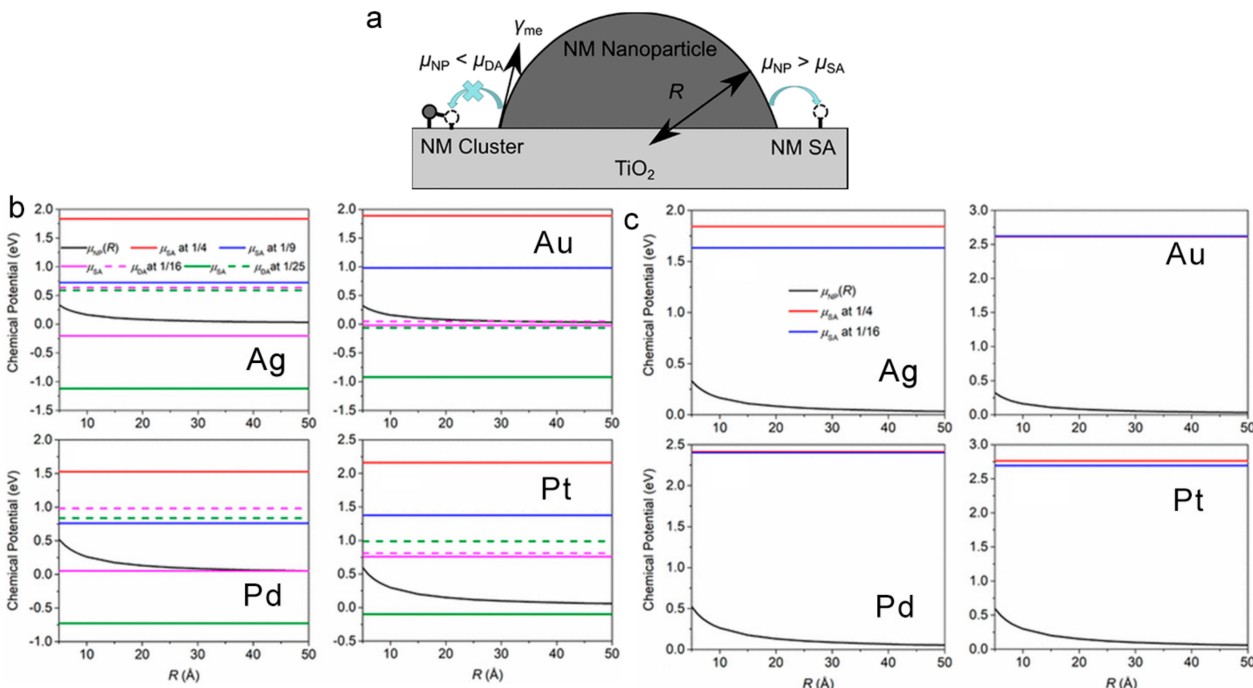

**Figure 5.** (**a**) Diagram of two constraint conditions on noble metal (NM) (NM = Ag, Au, Pd, and Pt) chemical potentials to prepare NM SAs on TiO$_2$ support, i.e., $\mu_{SA} < \mu_{NP} < \mu_{DA}$. (**b**) Chemical potentials of Ag, Au, Pd, and Pt SAs and DA clusters for the formation of NM-SA-(001) through compared with the chemical potentials of NM NPs as a function of their radii. (**c**) Chemical potentials of Ag, Au, Pd, and Pt SAs at two loading densities for the formation of NM-SA-(101) through comparing with the chemical potentials of NM NPs. Adopted with permission from Ref. [121].

As discussed above, to obtain stable metal SAs, they have to be stabilized on an appropriate support material by adequate trapping or covalent metal support interactions (CMSI). Again, such isolated metal atoms tend to aggregate forming small clusters and NPs due to the increased surface energy during synthesis and photo-reaction processes [122]. Since it is challenging to maintain the atomically isolated metal species, the successful formation of stable SACs strongly depends on the anchoring sites on the supporting semiconductor surface. In this regard, a surface defect engineering strategy was successfully applied to anchor and stabilize SACs on TiO$_2$ nanostructures [14,16,89,122–132]. The defect sites in the photocatalyst act as the anchoring points to trap single metal atoms. Self-organized TiO$_2$ nanotube (NT) layers grown by controlled anodic oxidation of a metal substrate have been explored for a wide range of functional applications and, in particular, for photo- and photoelectrocatalysis. Such high-aspect-ratio NT array architecture promotes efficient harvesting of photons by orthogonalizing the processes of light absorption and charge separation [133]. Here, Zhou et al. [134] utilized intrinsic defects in anodically grown TiO$_2$ NTs to anchor single Ir atoms. The formation of Ir SAs was confirmed by HAADF-STEM and EPR spectroscopies, while the uniformity of the SAC trapping was obtained from TEM-dispersive X-ray spectroscopy (Figure 6a). The annealed anatase NTs show significantly stronger defect signals and a distinctly different overall signature as compared to commercially available anatase TiO$_2$ NPs (Figure 6b). The EPR spectrum of TiO$_2$ nanotubes consists of response at $g \approx 2.0$ corresponding to Ti$^{3+}$ in the regular lattice position of TiO$_2$ and $g_{avg} \approx 1.9$ ascribed to surface-exposed Ti$^{3+}$-O$_V$ states. The latter is exclusively present in the NTs being highly effective to anchor metallic SAs. After interaction with Ir ions, the NT sample showed a significant decrease in the magnitude of the signature at $g_{avg} \approx 1.9$ suggesting that an attachment mechanism based on a galvanic displacement reaction, Ti$^{3+} \rightarrow$ Ti$^{4+}$ and Ir$^{3+} \rightarrow$ Ir$^{2+}$_{surface trapped}, occurs (Figure 6c). The XPS measurements revealed the existence of Ir SAC giving the oxidation state Ir$^{\Delta+}$. After 24 h

of irradiation of active sites, the turnover frequency (TOF) number for Ir SAC $TiO_2$ NTs reached $\sim 4 \times 10^6$ $h^{-1}$.

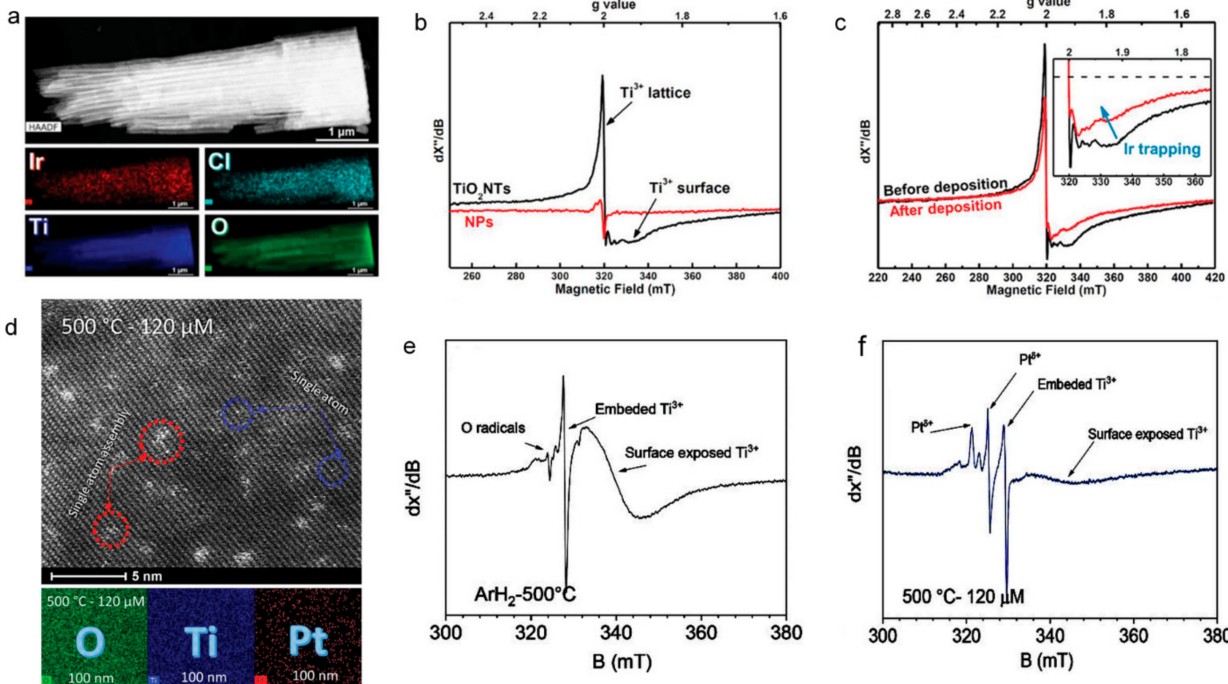

**Figure 6.** (**a**) TEM-EDX element mapping of $TiO_2$ nanotubes immersed in $IrCl_3$ solution for 24 h. (**b**) EPR spectra of stress-engineered $TiO_2$ NTs and anatase NPs, (**c**) EPR spectra of $TiO_2$ NTs before and after dark deposition of Ir SAs. (The inset shows the plots from $\approx$ 315–365 mT at higher magnification). Adopted with permission from Ref. [134]. (**d**) HAADF-TEM image and EDS map of Pt-SA-decorated $TiO_2$ layer. (**e**) EPR spectra of $TiO_2$ anatase annealed at 500 °C in $Ar/H_2$. (**f**) EPR spectra of the layer after immersion in the 120 µm hexachloroplatinic acid solution. Adopted with permission from Ref. [117].

Hejazi et al. [126] successfully loaded a single-atom Pt co-catalyst employing $Ti^{3+}$-$O_V$ defect states on $TiO_2$ nanostructures (Figure 6d). Here, the defect states were engineered by reducing $TiO_2$ nanostructures in $Ar/H_2$ environment. The EPR spectroscopy of regular lattice positions provides a typical signal at $g \approx 1.99$ and surface-exposed $Ti^{3+}$ shows a signal with a characteristic $g$ value of 1.93 (Figure 6e). After Pt SA loading on these defect sites, the EPR signal was reduced (Figure 6f). The formation of Pt single atoms was further confirmed by HAADF-STEM and XPS measurements revealing that the anchored single-atom Pt on $O_V$ has an oxidation state of $0 < \Delta^+ < 4$.

Wan et al. [135] investigated single Au atomic sites on defective $TiO_2$ nanosheets (Def-$TiO_2$). To obtain the defect states on $TiO_2$ support, the $TiO_2$ nanosheets were thermally treated at a reducing atmosphere ($H_2$:Ar) at different temperatures. According to the experimental results, $TiO_2$ with a high surface defect concentration of $O_V$ was generated at 200 °C under a reducing atmosphere. The structure of the Au site on Au-SA/Def-$TiO_2$ was analyzed by XPS, Au $L_3$-edge XANES, and extended X-ray absorption fine structure (EXAFS) measurements (Figure 7). Au 4f binding energy of Au-SA/Def-$TiO_2$ is 85 eV, which shifts to higher energy compared with pure Au NP (84 eV) (Figure 7a). The XANES curves of Au-SA/Def-$TiO_2$ showed the intensity of the white line peak closely resembles that of $HAuCl_4$ (Figure 7b). The EXAFS spectra (Figure 7c) showed the short-range (below 4 Å) local structure of Au in the as-prepared samples that one Au–O shell (R = 1.6 Å) and one Au–Ti shell (R = 2.4 Å) from Au-SA/Def-$TiO_2$ have been determined for atomic Au site without Au–Au and Au–Cl coordination peaks, compared with Au foil and $HAuCl_4$. A least-squares EXAFS fitting demonstrated quantitative structural parameters of Au in

Au-SA/Def-TiO$_2$. The coordination number of Au–Ti on Au-SA/Def-TiO$_2$ was 1.8 and the mean bond length was 2.79 Å. Based on the above analysis, the authors concluded that the surface defects on TiO$_2$ nanosheets can stabilize Au SAs by constructing a three-center Ti–Au–Ti structure and this can promote the catalytic properties by reducing the energy barrier and relieving the competitive adsorption on isolated Au atomic sites.

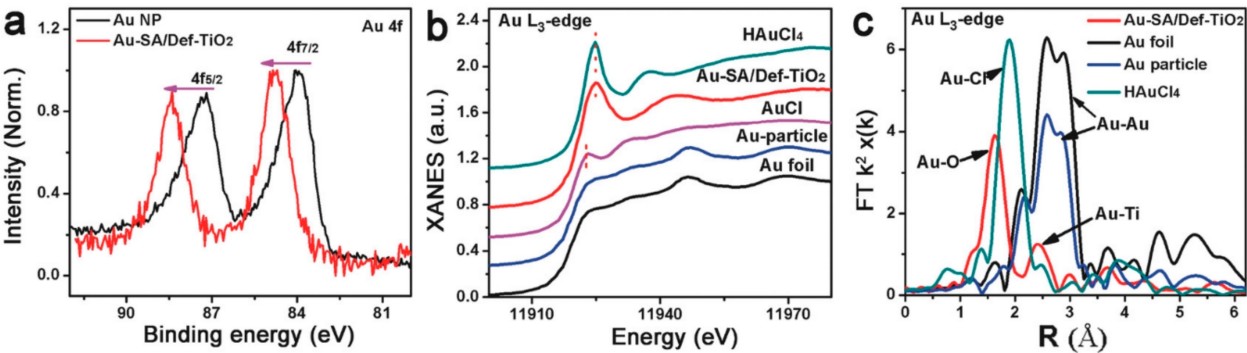

**Figure 7.** XPS and EXAFS analysis of Au-SA/Def-TiO$_2$: (**a**) Au 4f XPS spectra of Au-SA/Def-TiO$_2$. (**b**) XANES spectra. (**c**) Fourier transform (FT) of the Au L$_3$-edge. Adopted with permission from Ref. [135].

The organometallic chemistry approach was also applied to form SA catalysts. For instance, Fan et al. [136] constructed single atom Ni sites on TiO$_2$ using the surface organometallic chemistry of nickelocene for "p–n heterojunction"-induced visible-light photocatalysis. The surface hydroxyls of anatase react with nickelocene to form a surface CpNi –O-Ti≡ complex. Upon calcination in oxygen, the surface complex transforms into a surface nickel-oxo species, where Ni is bonded to two lattice oxygen atoms of TiO$_2$ and four oxygen atoms from adsorbed water molecules with a Ni-O distance of 2.01 Å, where all nickel atoms are present in the form of $(\equiv TiO)_2Ni(H_2O)_4$. According to the authors, these species create small "*p*-type zones", strongly affecting the local electric structure of neighboring TiO$_6$ octahedra. The characterization results indicated that atomically isolated Ni sites on TiO$_2$ were responsible for the visible light absorption and photocatalysis of Ni/TiO$_2$ composites. Gu et al. [137] prepared Sn-modified TiO$_2$ by the grafting reaction of tetramethyltin. Tetramethyltin was preferentially grafted on the Ti$^{3+}$–OH sites at the (101) and (001) planes of TiO$_2$. After calcination in oxygen, tin was atomically bound to TiO$_2$ forming the Ti$^{IV}$–O–Sn$^{IV}$ heterobinuclear clusters.

Density functional theory (DFT) is a vital tool to explain the configuration of SA on TiO$_2$ surfaces. Chen et al. [138] used DFT calculations to investigate the adsorption of Cu SAs over stoichiometric and reduced (101) surface of TiO$_2$ (Figure 8a). Over the stoichiometric surface, the Cu atom preferred a bridge geometry between two O$_{2c}$ atoms, with an adsorption energy of −2.30 eV (Figure 8b). Over the reduced surface, three stable sites are possible. Cu atom tends to occupy oxygen vacancy, with an adsorption energy of −2.23 eV (Figure 8c). Slightly lower adsorption energy (−2.12 eV) was found when the Cu atom adsorbed near the vacancy but did not completely occupy it (Figure 8d). When the Cu atom was far from the oxygen vacancy, it similarly interacted with the surface as with the stoichiometric surface, giving comparable adsorption energy (−2.26 eV) (Figure 8e). The Bader charge calculations reveal that the Cu atoms became positively charged when interacting with O$_{2c}$ atoms (stoichiometric and far from the oxygen vacancy) but negatively charged when in or near the oxygen vacancy. The region around the vacancy is negatively charged due to unpaired electrons left on the surface on nearby Ti atoms upon O removal, and the surface donates this charge to the adsorbed Cu atom. Bader charge also reveals that Cu SA is stabilized on TiO$_2$ via electronic metal-support interaction (EMSI).

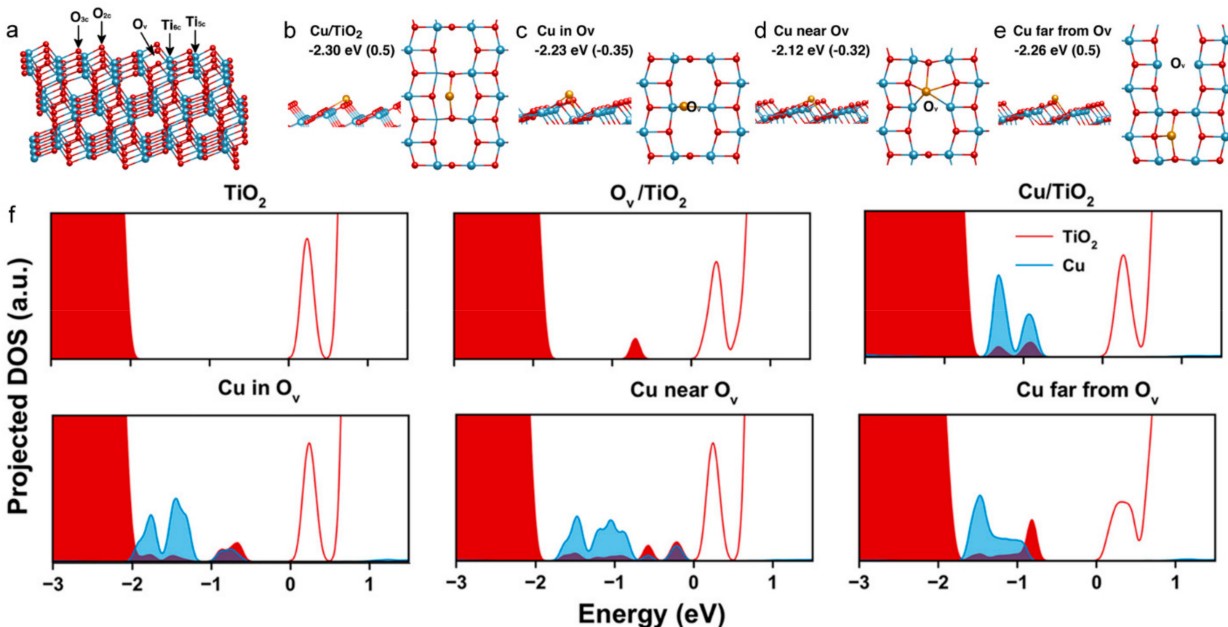

**Figure 8.** (**a**) Supercell—a (2 × 4) representation of the anatase (101) surface. Indicated are different atom types and an oxygen vacancy ($O_v$). The subscripts indicate the coordination number of the atoms, e.g., $O_{2c}$ is a two-coordinated oxygen atom. $O_V$ indicates an oxygen vacancy. (**b**–**e**) Cu adsorption over $TiO_2$ anatase (101) surfaces. Cu adsorption (**b**) over a stoichiometric surface, (**c**) in an oxygen vacancy, (**d**) near an oxygen vacancy, and (**e**) far from the oxygen vacancy. The Cu atom is depicted in orange. Given are adsorption energies while numbers in parentheses are calculated Cu charges. (**f**) Projected density of states plots for adsorbed Cu. The zero-energy level is set at the conduction band edge. The filled curves represent occupied states, while unfilled curves represent unoccupied states. Adopted with permission from Ref. [138].

Figure 8f shows the density of states (DOS) for Cu adsorbed on the various $TiO_2$ surfaces. DOS for stoichiometric and reduced $TiO_2$ surfaces are given for comparison. A gap state for a surface with $O_V$ was observed due to electrons localizing on Ti atoms. For Cu over $TiO_2$, a gap state created by Cu was also observed. For Cu in an $O_V$, a similar gap state due to Cu occurs (near −1.5 eV), but there is also a gap state on $TiO_2$ due to the $O_V$ near −0.7 eV, which is composed of Ti electrons. Overlap between the Cu and $TiO_2$ gap states suggests strong interactions, such as Cu in an $O_V$ (−0.7 eV) and Cu near an $O_V$ (between −0.4 to 0 eV). Indeed, electron transfer occurs from the $O_V$ to Cu, with Cu having a −0.35 $e^-$ charge in the $O_V$. When an $O_V$ forms, two unpaired electrons are left in $TiO_2$ ($O^{2-} \rightarrow O_v + 1/2O_2 + 2e^-$), which may reduce two Ti atoms. The two reduced Ti atoms near the $O_V$ have charges of 1.93 and 1.83 $e^-$, compared to the average charge of 2.42 $e^-$ for the other Ti atoms. After Cu was adsorbed in an $O_V$, one of the reduced Ti atoms had a charge of 2.19 $e^-$, while the other reduced Ti atom had a charge of 1.73 $e^-$, indicating a net transfer of electrons to the Cu atom from the two Ti atoms. After the charge transfer occurred from $TiO_2$ to Cu, the $O_V$ gap states hybridized with the Cu states. For Cu far from an $O_V$, both gap states for Cu and $TiO_2$ are seen, but since the Cu does not interact with the $O_V$, these states are largely independent of each other and have little overlap. Zhang et al. [87] also estimated that Cu preferentially sits on the bridge-center site between two 2-fold coordinated O atoms ($O_{2c}$), while Cu and coordinated $O_{2c}$ act as copper oxide species (-Cu-O-) in the reaction.

## 5. Other Aspects of Single Atom Co-Catalysts in Photocatalysis

### 5.1. Light Absorption/Harvesting

Light (photons with energies equivalent to or higher than the bandgap energy of the photocatalyst) absorption or harvesting is the first fundamental step in photocatalysis. Semiconducting materials are used as a catalyst to perform sensitization of light stimulating redox processes due to their electronic structure, i.e., a filled valence band and a vacant conduction band [139]. However, loading of a single atom on a host semiconductor may emerge new electronic states in the bandgap leading to the enhancement of solar light absorbance. Li et al. [140] demonstrated the effect of $Ru_1$ single-atom decorated monolayered $TiO_2$ nanosheets (TiNS). While the UV-Vis diffuse reflectance spectroscopy of TiNS showed narrow optical absorbance with an edge at $\lambda = 360$ nm, attributed to the charge transfer from O $2p$ for valence band to Ti $3d$ for conduction band, the $Ru_1$/TiNS sample displayed a broad light absorption range up to $\lambda = 700$ nm (Figure 9a). This was attributed to an additional charge transfer route from Ru $d$-$d$ transition (shoulder peak around 470 nm) and charge transfer from VB of Ru $4d$ or O $2p$ to oxygen vacancy (between 360 nm and 470 nm), as oxygen vacancy has an energy level located 0.75–1.18 eV below the conduction band. Photocurrent density for $Ru_1$/TiNS was also higher than that of plain TiNS, suggesting more efficient charge separation after photoactivation as well as additional electrons generated from Ru energy level to CB (Figure 9a, inset image). Using experimental characterization and theoretical calculations, the authors demonstrated that both $Ru_1$ dopant and the induced oxygen vacancies play key roles in promoting visible light driving the $H_2$ evolution reaction. Single-atom $Ru_1$ replacing Ti induces an isolated impurity energy level between VB (O $2p$) and CB (Ti $3d$), which extends the light absorption range up to $\lambda = 700$ nm, lowering the $H_2$ evolution reaction barrier by promoting $H^+$ adsorption at the O site from Ru-O-Ti. Furthermore, $O_V$ created by the introduction of Ru dopants extends light absorption from $\lambda = 360$ nm to 470 nm, serving as an electron trapping site promoting photogenerated electron separation and transportation. Similarly, Niu et al. [42] demonstrated the absorption spectra of a Ru-based single-atom catalyst anchored on defect-rich $TiO_2$ nanotubes (Def-NTs). The absorption spectrum of the Ru-SAs/Def-TNs exhibited a protruding drum peak at $\sim$460 nm ($\sim$2.70 eV) corresponding to a ligand (Def-TNs) to metal center ($Ru^{\Delta+}$) charge-transfer (LMCT) state as a result of the polar bond produced by the $Ru^{\Delta+}$–Ox coordination, bringing on the spatial redistribution of electrons between the oxidized $Ru^{\Delta+}$ metal and the reduced Def-TNs support (caused by the introduced oxygen-vacancy defects) (Figure 9b) [141–143]. In contrast, the spectral feature of the LMCT state was not observed in Ru-NPs/Def-TNs showing only a flat, weak absorption across the visible region of the spectrum (caused by the scattering of light by Ru-NPs).

Wang et al. [144] studied theoretically the (Rh + F) surface co-doping effect on anatase $TiO_2$ (101) by density functional theory. The noble Rh metal atoms can be stably doped at the anatase $TiO_2$ (101) surface with the aid of the co-doped F. To assess the surface co-doping effect, the total density of states (DOS) of the (Rh + F) doped anatase $TiO_2$ (101) surface and the partial DOS of the O $2p$, Ti $3d$, F $2p$, and Rh $4d$ orbitals were calculated and compared with the corresponding results for the pure $TiO_2$ surface (Figure 9b). In contrast to the Rh monodoping case, the ground state of the (Rh + F) doped $TiO_2$ surface is spin-restricted, and the magnetic moment is zero, indicating the absence of an unpaired electron in the co-doped system. Figure 9b demonstrates that the (Rh + F) co-doping pair just slightly perturbs the VBM and CBM positions compared to the pure anatase $TiO_2$ surface, while intermediate bands located below the Fermi level appear within the bandgap. The effective bandgap of the (Rh + F) co-doped $TiO_2$ surface was narrowed to about 2.14 eV. According to the calculated partial DOS, these occupied intermediate bands are mainly contributed by O $2p$, Rh $4d$, and F $2p$ orbitals (Figure 9d). Such broad, delocalized, and occupied intermediate bands induced by the (Rh + F) co-doping pair reduce the bandgap but can also prevent the recombination of photogenerated carriers. Moreover, no acceptor states appear above the Fermi level indicating that the recombination

centers are greatly reduced compared to the Rh or F monodoping cases. The (Rh + F) co-doped anatase TiO$_2$ (101) surface demonstrates a significant redshift of the absorption edge along both parallel and vertical directions relative to the anatase TiO$_2$ (101) surfaces (Figure 9c). Such a reduction of the bandgap of the (Rh + F) co-doped anatase TiO$_2$ (101) surface can potentially harvest the light also in the visible spectral range. It should be noted, however, that the above examples demonstrate the possible enhancement of light absorption based mainly on DFT calculations and have not been confirmed experimentally. While the absorbance spectra between plain and SAC-decorated samples appear in a few publications only, in the majority of publications, the comparison of the absorbance spectra between these samples either is not shown or no obvious difference was observed, yet SAC-decorated samples displayed significantly enhanced photocatalytic performance as compared to their plain counterparts [145]. Therefore, more evidence is required to clarify this highly important point.

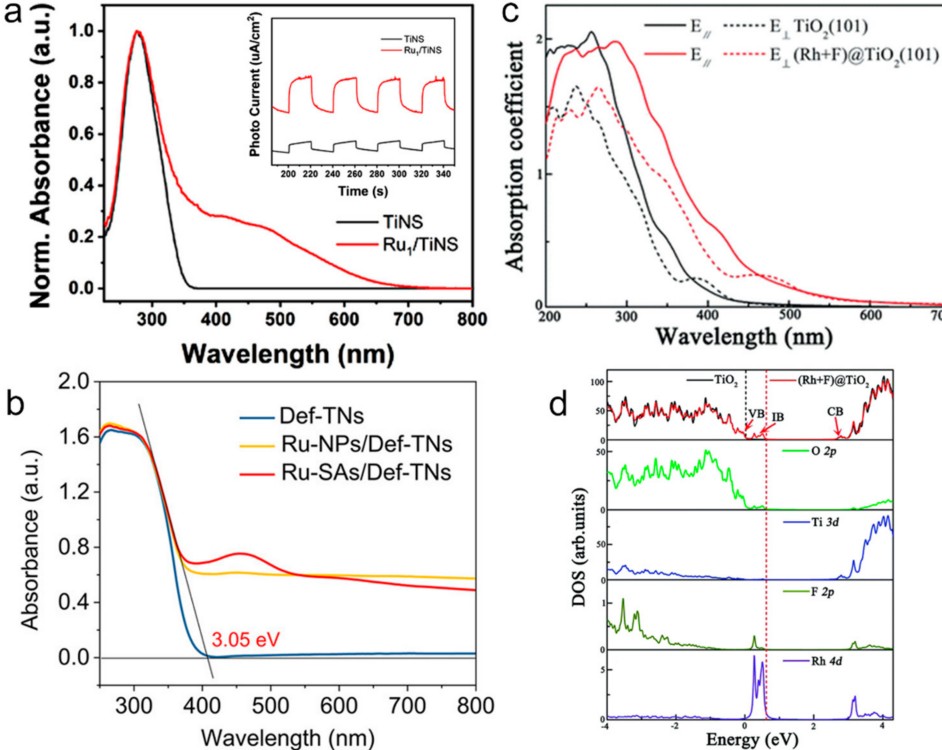

**Figure 9.** (**a**) UV–vis DRS of TiNS and Ru$_1$/TiNS. Adapted with permission from Ref. [140] (**b**) UV-vis absorption spectra. Adopted with permission from Ref. [42] (**c**) Calculated optical absorption coefficients of the pure and (Rh + F) co-doped anatase TiO$_2$ (101) surfaces. ∥ and ⊥ represent parallel and perpendicular directions relative to the (101) surface. (**d**) Calculated total DOS of the (Rh + F) co-doped anatase TiO$_2$ (101) surface and partial DOS of the O, Ti, F, and Rh atoms in the co-doped system, respectively. The black line shows the total DOS of the pure anatase TiO$_2$ (101) surface for clarity, and the black and red vertical dashed lines represent the Fermi levels for the pure and co-doped systems, respectively. Adapted with permission from Ref. [144].

## 5.2. Single-Atoms and Local Electronic Structure

When the size of the nanocrystal is reduced to the form of clusters of atoms and finally to single atoms, its energy levels, and electronic structure alternate [146], i.e., SACs can modulate the electronic structure locally to enhance photocatalytic activity, meaning that the adsorbed metal atoms induce new energy states in the bandgap of the semiconductor acting as an electron trapping center for the photogenerated electrons enhancing charge transfer between light-harvesting material and single-atoms. Hence, it is highly demanding to obtain an insightful understanding of the structural geometries of SAs and the electronic

interactions between SAs and oxide substrates, as well as the responses of SAs under light illumination, which are crucial for further understanding their synergistic catalytic mechanisms. Dong et al. [147] investigated the response of Au SAs on the TiO$_2$ (110) surface under UV light illumination. DFT calculations showed the existence of the metal-induced gap states (MIGSs) for Au SAs at the Ti$_{5c}$ and O$_V$ sites. In the case of metal clusters on oxide surfaces, their electronic interaction is initiated by the interfacial charge transfer resulting from the balancing of the Fermi level ($E_F$) of the two kinds of materials [148]. Commonly, the Fermi level pinned at the metal/oxide interface makes the metal electronic states extend into the oxide bandgap leading to the formation of MIGSs [149,150]. Similar to the gap states introduced by the intrinsic defects in TiO$_2$ [151], the MIGS can provide excess charges at the interface, inducing higher catalytic activity at the interface. In the SAs, the existence of the MIGS can be similar and even more important because the MIGS originates from the interaction between the SAs and the TiO$_2$ substrate providing a potential channel for the charge transfer for the photoexcited carriers. The Au SAs present site-specific adsorption characteristics at the O$_V$ and the Ti$_{5c}$ sites. Under UV light illumination, the Au SAs have a high possibility to move between the Ti$_{5c}$ sites and even falling into the O$_V$ sites, while the Au SAs adsorbed at the O$_V$ sites do not tend to move out, i.e., the Ti$_{5c}$-bonded Au SAs present a strongly localized MIGS at −1.2 eV below $E_F$. Under UV light illumination, the photoexcited hole in the TiO$_2$ valence band transfers to the Au SAs through the MIGS weakening the Ti-Au bonding, which activates the diffusion of Au SA across the surface. The localized MIGS below the Fermi level provides a dedicated channel for the transfer of photoexcited holes from the TiO$_2$ substrate to the adsorbed Au SAs, i.e., are responsible for the hole-induced diffusion.

Han et al. [121] used the first principle approach to investigate charge transfer during photoexcitation. To simulate the charge transfer, an extra electron was added to the system. The difference charge density calculated via the generalized gradient approximation of the Perdew–Burke–Ernzerhof with the unrestricted Hartree–Fock approximation showed that the photoelectron is trapped at Ti$_{6c}$ in the subsurface of (001) TiO$_2$. Trapping of a photoelectron at Ti$_{6c}$ sites is the reason for the high energy barrier for the surface reactivity, i.e., this energy barrier must be overcome upon the charge migration to the surface that suppresses the photoinduced reactivity of the plain (001) TiO$_2$ plain. After noble metal (NM) deposition, excess electrons are mainly distributed around NM and O. In addition, DOS calculations reveal that the conduction band of plain (001) consists of subsurface Ti$_{6c}$ states rather than Ti$_{5c}$ explaining the trapping of photoelectrons in the (001) subsurface (Figure 10a). After the deposition of NM SAs, unoccupied states were observed in the band gap between the highest occupied states and conduction band primarily contributed by hybridization of NM and Ti$_{5c}$, which act as electron acceptor (Figure 10b). This was verified by the Bader charge indicating that NM SAs are positively charged and electrons are transferred from NM SAs to TiO$_2$. Upon photoexcitation, excess electrons are trapped around SAs, as evidenced by the negative Bader charge of NM SA. (Bader charge before photoelectron trapped: 0.39, 0.01, 0.15, and 0.02 for Ag, Au, Pd, and Pt, and after photoelectron trapped: −0.42, −0.35, −0.26, and −0.34 for Ag, Au, Pd, and Pt, respectively). NM SAs on the (001) facet can pump electrons to the catalyst surface, facilitating the subsequent proton adsorption and electron transport process of HER. When SAs are anchored on TiO$_2$, the metal support interaction enables the transfer of an electron from SAs to TiO$_2$ support. However, the electron is transferred from TiO$_2$ to SA in case of defective TiO$_2$. Such an interaction could modify the conduction and valence band level of TiO$_2$, thereby changing its oxidizing and reducing capabilities, i.e., the reducing capacity of photocatalyst increases in the case of a more positive conduction band level, while the more negative valence band increases the oxidizing capacity [20]. In the case of pure TiO$_2$, the conduction band is just above the redox potential of H$_2$O (Figure 10c). After the adsorption of metal atoms on anatase TiO$_2$ (101) surface, the conduction band shifts upward increasing its reducing capacity. Meanwhile, the upward shift in the valence band of TiO$_2$ reduces its oxidizing capacity. Furthermore, the change in potential of the topmost Ti atom was

calculated for its role in the photocatalytic process. As shown in Figure 10d, the PDOS of the topmost Ti 3d band for the single metal atoms adsorbed anatase TiO$_2$ surface shifted toward the vacuum level, indicating the higher reduction ability of the Ti atoms. Therefore, in addition to adsorbed metal atoms, the surface Ti atoms serve as reducing sites for the photocatalytic process.

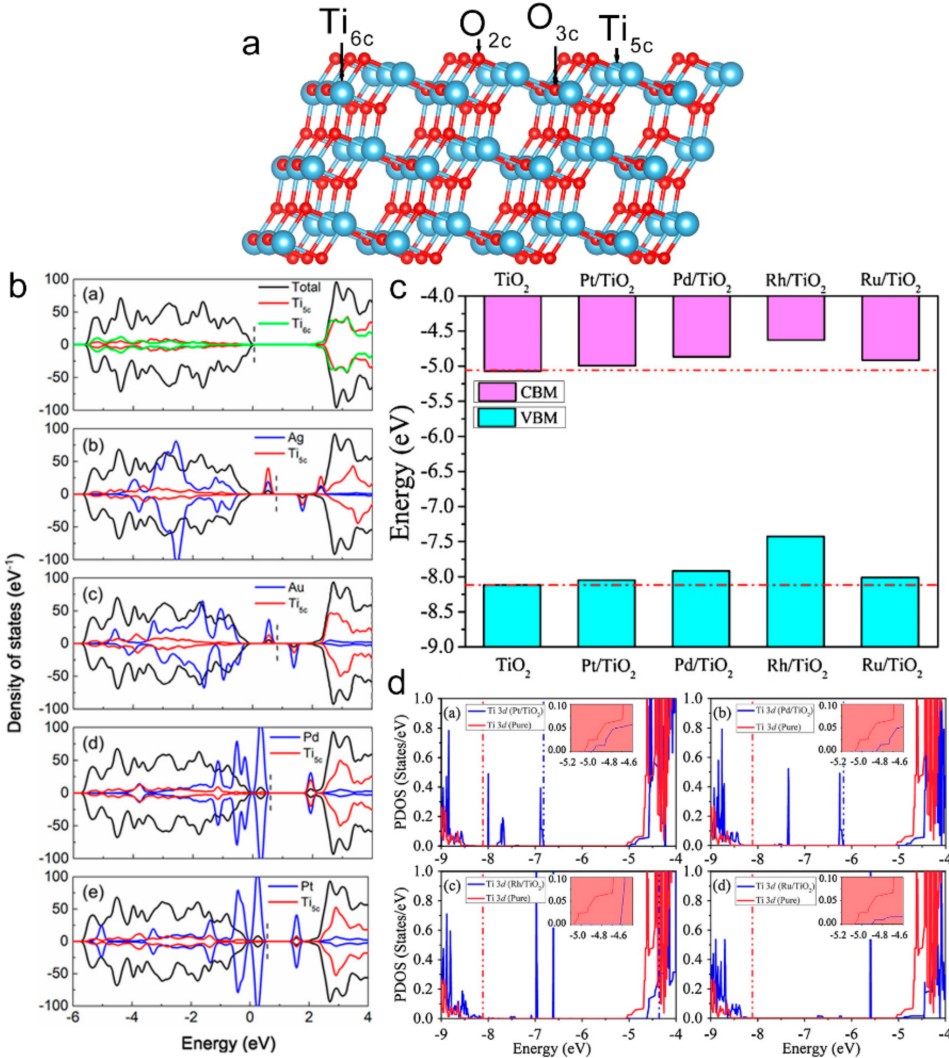

**Figure 10.** (**a**) Optimized structure of the plain anatase TiO$_2$ (101) surface. (**b**) Total and local DOS calculated with PBE + *U* for plain (001), Ag-, Au-, Pd-, and Pt-SA-(001) at 1/9 loading density. Ti$_{5c}$ and Ti$_{6c}$ of the panel a represent the DOSs contributed by surface 5-coordinated Ti atoms and subsurface 6-coordinated Ti atoms, respectively. For clarity, the DOSs of NM and Ti5c of panels b–e are magnified by 20 times, and all the TiO$_2$ valence bands which are mainly composed of O 2*p* states are aligned roughly to 0 eV. The vertical short dashed line in each panel denotes the highest occupied state in each case. Adopted with permission from Ref. [121] (**c**) Change of conduction band minimum (CBM) and valence band maximum (VBM) of the host TiO$_2$ upon the single Pt, Pd, Rh, and Ru atom adsorption on anatase TiO$_2$ (101) surface, with the vacuum level set to zero. (**d**) Calculated partial density of states (PDOS) of the topmost surface Ti 3*d* orbits for single Pt, Pd, Rh, and Ru atom adsorbed anatase TiO$_2$ (101) surface. The vacuum energy level is set to zero. The insets show the band edge of the conduction band, and the vertical dashed lines represent the Fermi level. (**a**,**c**,**d**) Adopted with permission from Ref. [20].

## 6. Applications of Single-Atom Co-Catalyst in Photocatalysis

### 6.1. Hydrogen Production

Since the discovery of hydrogen photocatalytic generation by Fujishima and Honda in 1972 using $TiO_2$ as a photocatalyst (photoelectrode), this clean energy vector has received extensive attention [152]. However, a main bottleneck of photocatalysis, as was discussed above, is the sluggish reactions of common photocatalytic materials that require co-catalysts. Here, SACs have the potential to enhance the PC performance of photocatalysts owing to the better utilization of metal co-catalyst atoms with higher photocatalytic activity. For instance, Hejazi et al. [126] developed a tunable strategy combining surface defect engineering and decoration of Pt SAs. The key trap for the Pt is surface $Ti^{3+}$-$O_V$ with a characteristic *g* value of $\approx 1.93$ in EPR measurements. Such surface-exposed $Ti^{3+}$ defects can be formed in a highly controlled manner by reducing thin $TiO_2$ layers in an $Ar/H_2$ environment at different temperatures. Subsequently, these atomic-scale defective sites were used to pin single Pt atoms from a dilute aqueous Pt precursor solution. HAADF-STEM results confirm the SA decoration of the $TiO_2$ surface with Pt atoms. The amount of Pt deposition can be varied from 0.03 to 0.47 at.% by varying the temperature of the heat treatment in $Ar/H_2$. The XPS and EPR results confirmed the contribution of $Ti^{3+}$ states in the deposition of Pt atoms. The SA-decorated $TiO_2$ samples demonstrated a 150-times increase in the specific $H_2$ evolution rate as compared with the $H_2$ generation rate of the $TiO_2$ layer decorated by classic Pt NPs.

Liu et al. [89] stabilized Pt SAs on reduced $TiO_2$ (Pt SA/Def-$TiO_2$) using oxygen vacancies and measured their HER photocatalytic activity. The presence of oxygen vacancies (defects) was confirmed by EPR demonstrating the *g*-value response at 2.003. After Pt loading, the EPR signal decreased indicating that the Pt was mainly anchored at the $O_V$ sites, which was further confirmed by HAADF-STEM and XPS measurements (Figure 11a,b). The Pt 4f XPS signal was shifted to higher binding energy compared to $Pt^0$ indicating that Pt exists as $Pt^{\Delta+}$. The $H_2$ evolution of Pt SA/Def-$TiO_2$ reached 13.46 mmol $h^{-1}$ $g^{-1}$ (loading 0.57 wt.%), which was ~5-fold higher than that of Pt NP/s-$TiO_2$ counterparts. The higher activity of Pt SA-based photocatalyst was attributed to the H* adsorption free energy being closer to the optimal value of 0 as compared to Pt NP/$TiO_2$, as well as to the improved Pt atom utilization efficiency. Chen et al. [153] prepared Pt SA on sodium titanate followed by calcination in the reduced atmosphere at 160 °C to form $Pt_1$/Def-$TiO_2$. Apart from being proton reduction sites, Pt SAs promoted the neighboring $TiO_2$ units to generate surface oxygen vacancies forming a Pt-O-$Ti^{3+}$ atomic interface that facilitates photo-generated electrons to transfer from $Ti^{3+}$ defective sites to Pt SAs, thereby, enhancing the separation of electron-hole pairs. The $Pt_1$/Def-$TiO_2$ photocatalyst achieved a turnover frequency (TOF) of $5.14 \times 10^5$ $h^{-1}$, which was ~600-fold higher than that of Pt-NPs-supported $TiO_2$ counterparts, and also showed a small decline in continuous $H_2$ production during the subsequent cyclic runs (Figure 11a,b).

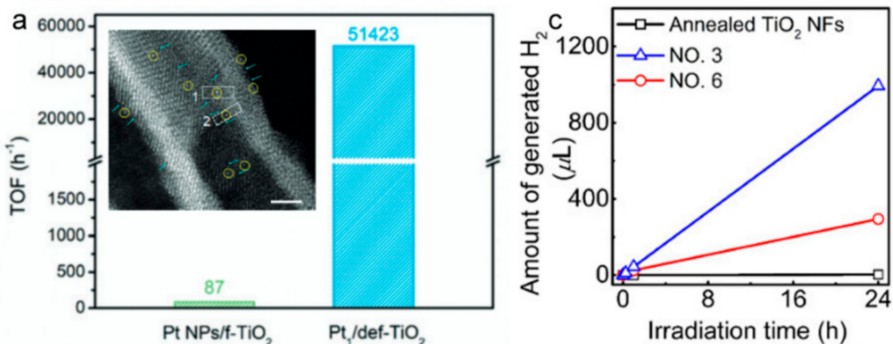

**Figure 11.** *Cont.*

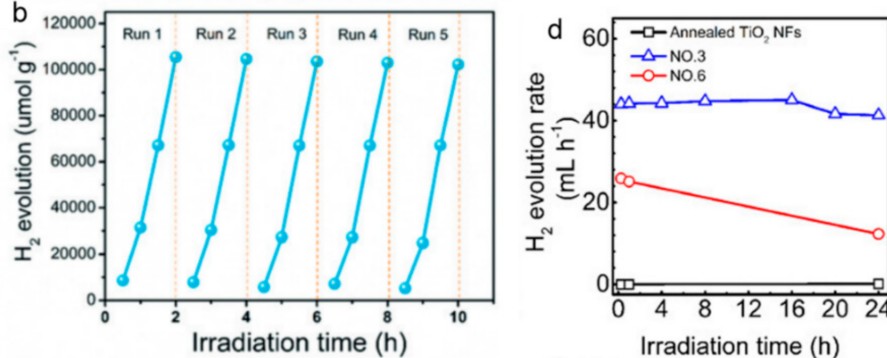

**Figure 11.** (**a**,**b**) The TOFs photocatalytic $H_2$ production activities and cyclic tests under UV/Vis light illumination of Pt-SA/Def-TiO$_2$ and Pt-NP/TiO$_2$. Inset in (**a**) is the TEM image of Pt-SA/Def-TiO$_2$. The scale bar is 2 nm. Adapted with permission from Ref. [153] (**c**) Amount of $H_2$ evolution photogenerated over 24 h of irradiation at $\lambda$ = 365 nm for the annealed, the "dark"-deposited (NO. 3), and the pre-photodeposited (NO. 6) TiO$_2$ NFs. (**d**) Photocatalytic $H_2$ evolution rate over 24 h illumination for the samples shown in (**c**). Adapted with permission from Ref. [145].

Cha et al. [145] decorated Pt SAs on the (001) plane of anatase TiO$_2$ nanoflakes (NFs) via immersion, i.e., a "dark"-deposition, technique. The anchoring sites were formed by annealing TiO$_2$ NFs at 450 °C in air. Pt SA-based photocatalyst achieved 4-fold higher photoreactivity than Pt NP-decorated TiO$_2$ NFs (Figure 11c,d). Moreover, in this case, the Pt SA demonstrated a stable $H_2$ production rate for 24 h. Qin et al. [154] investigated the co-catalytic activity of Pt SA on different polymorphs of TiO$_2$ NPs such as anatase, rutile, and mixed-phase P25 for water splitting. The Pt SAs were incorporated into TiO$_2$ NPs by the immersion technique. For Pt SA-loaded rutile NPs, the increase in co-catalytic activity was not observed, while the Pt SA decorated anatase and mixed-phase P25 NPs showed a clear increase in co-catalytic activity also as compared to P25 decorated by Pt NPs. Moreover, in the case of P25, a minute amount of Pt SAs provided a pronounced increase of co-catalytic efficiency, i.e., SA density of $5.3 \times 10^5$ $\mu m^{-2}$ displayed an $H_2$ production rate of 4600 $\mu mol\ g^{-1}\ h^{-1}$. Wu et al. [127] studied anodic TiO$_2$ nanotubes (NT) decorated with Pt SAs for $H_2$ production. The Pt SAs were anchored on NTs by the "dark"-deposition immersion technique. To trap Pt SAs, the optimal anchoring sites Ti$^{3+}$-O$_V$ were created by annealing in air due to the crystallization process of NTs. The formation of defects was evaluated by EPR measurements demonstrating that NTs annealed at 350 °C displayed an isotropic signal only corresponding to bulk defects, while an additional axial signal (corresponding to surface Ti$^{3+}$-O$_V$ defects) was observed in TiO$_2$ NTs annealed at 550 °C and 750 °C, while 550 °C was found to be optimal. The resulting SA decorated TiO$_2$ NTs provide co-catalytic performance with TOF for $H_2$ generation of $1.24 \times 10^6$ $h^{-1}$ at a density of $1.4 \times 10^5$ Pt atoms $\mu m^{-2}$. Furthermore, the Pt SA decorated TiO$_2$ NTs demonstrated stable photocatalytic activity over an illumination time of 24 h. Wang et al. [155] explored the photocatalytic activity of Pt SAs on a self-assembled monolayer (SAM) modified TiO$_2$ NTs. To anchor Pt SA, anodized and annealed TiO$_2$ layers were treated with an organic monolayer using silane chemistry followed by cutting the hydrocarbon tail. The SAM-modified surface achieved a 3-fold enhancement photo-catalytic $H_2$ production rate as compared to unmodified TiO$_2$ for the same Pt loading.

Cha et al. [14] investigated the noble metals, such as Pt, Pd, and Au, SAs for photo-catalytic HER. The SAs were anchored on {001}-faceted anatase TiO$_2$ nanosheets through surface vacancies (Ti$^{3+}$-O$_V$) introduced by Ar/H$_2$ thermal reduction treatment. XPS measurements were carried out to elucidate the coordination state and charge of the noble metal SAs. The XPS spectra revealed that Pt and Pd exist as SA in the form of Pt$^{\Delta+}$ (80%) and Pd$^{\Delta+}$ (70%), while Au exists as Au$^{\Delta+}$ (55%). The authors found that Pd SAs radically outperform Pt and Au SAs deviating from the volcano-type relationship in the case of NPs, i.e., the $H_2$ evolution reaction activity sequence is Pt > Pd > Au (Figure 12a–c) [86]. To

elucidate such unpredicted photocatalytic $H_2$ evolution activity of the Pd SA in comparison to SAs of Pt and Au on $TiO_2$, DFT calculations were performed. The obtained results demonstrated that the incorporation of the metal SAs into the $O_V$ site is energetically unfavorable to the build-up of the pure metallic phase as metal adsorption energies are below the cohesive energies of all three studied metals. The weakest interaction was found for Au ($E_{ads}$(Au) = −2.04 eV), while $O_V$ sites serve as binding sites at low SA concentrations. The authors further investigated hydrogen interaction with the SA sites expecting that the overall trend follows volcano-type performance. In fact, $E_{ads}$(H) follows the trend of the observed photocatalytic activity with Pd binding H with −3.46 eV, Pt with −3.22 eV, and Au with only −2.04 eV. With the addition of the second H to the SA sites, the situation became even more complicated, as this trend was lost. However, when mobile photoelectrons are generated, they should go to the reactive sites, i.e., the SA ones, to generate $H_2$. To model this process, one extra electron was added to the H/SA-$TiO_2$ and 2H/SA-$TiO_2$ models. In the latter case, the charge localization was the most prominent for Pd SA indicating that effective photoelectrocatalysis by metal SA requires both appropriate H binding energetics and proper charge localization of photoelectrons by co-catalyst SA sites.

Qin et al. [154] successfully decorated Pt SAs on pure anatase, pure rutile, and mixed-phase (P25) $TiO_2$ polymorphs by an immersion technique without any post-annealing or other subsequent treatments of the samples. This immersion approach leads to oxygen-coordinated Pt SAs (with δ+ ≈ 2) and an activity of the Pt SA decorated P25 and anatase polymorphs reaching 4600 μmol $g^{-1}$ $h^{-1}$ for Pt SA density of ≈5.3 × $10^5$ $μm^{-2}$ on P25. This is a remarkable activity of the photocatalyst considering that the total loading of Pt on this sample is only 0.06 at.%. If compared to the conventional nanoparticulate Pt co-catalysts, Pt SA loading yields a higher $H_2$ evolution performance and significantly higher Pt utilization. Based on the above observations, one may deduce that (i) a minute amount of Pt SAs is able to provide a very high degree of co-catalytic efficiency on P25, (ii) for higher Pt SA concentrations, the recombination reaction of $H_2 + O_2 \rightarrow H_2O$ affects the $H_2$ production detrimentally, (iii) the beneficial effect of P25 is due to its embedded junction. This effect particularly is relevant (in comparison to anatase), if the Pt SA surface density is low. At the same time, Pt SA loading on rutile has hardly any effect on the $H_2$ production rate. These results showes that not the presence of SAs is the main factor to enhance the photocatalytic $H_2$ production on the different titania polymorphs but its most efficient effect is the suppression of charge carrier recombination if relevant surface states are present.

Zhou et al. [134] prepared Ir SAs exploiting intrinsic defects in anodically grown oxide nanotubes as anchoring binding sites (Figure 12d). In self-organized anodic $TiO_2$ nanotubes, a high degree of stress is incorporated into the amorphous oxide during nanotube growth. During crystallization (by thermal annealing), this leads to a high density of $Ti^{3+}$-$O_V$ surface defects that are hardly present in other common $TiO_2$ nanostructures being highly effective for SA Ir trapping. The photocatalytic activity of such Ir SA on $TiO_2$ NTs was higher than SAs on other $TiO_2$ nanostructures, i.e., nanostructures with a higher surface area or geometries that are facet-engineered but also for conventional catalyst-NP decoration on $TiO_2$ nanostructures (Figure 12e). The high photocatalytic activity was attributed to the unique surface defects of anodically grown $TiO_2$ NTs, while such Ir SAs stably trapped on $TiO_2$ NTs can reach TOFs of ~4 × $10^6$ $h^{-1}$.

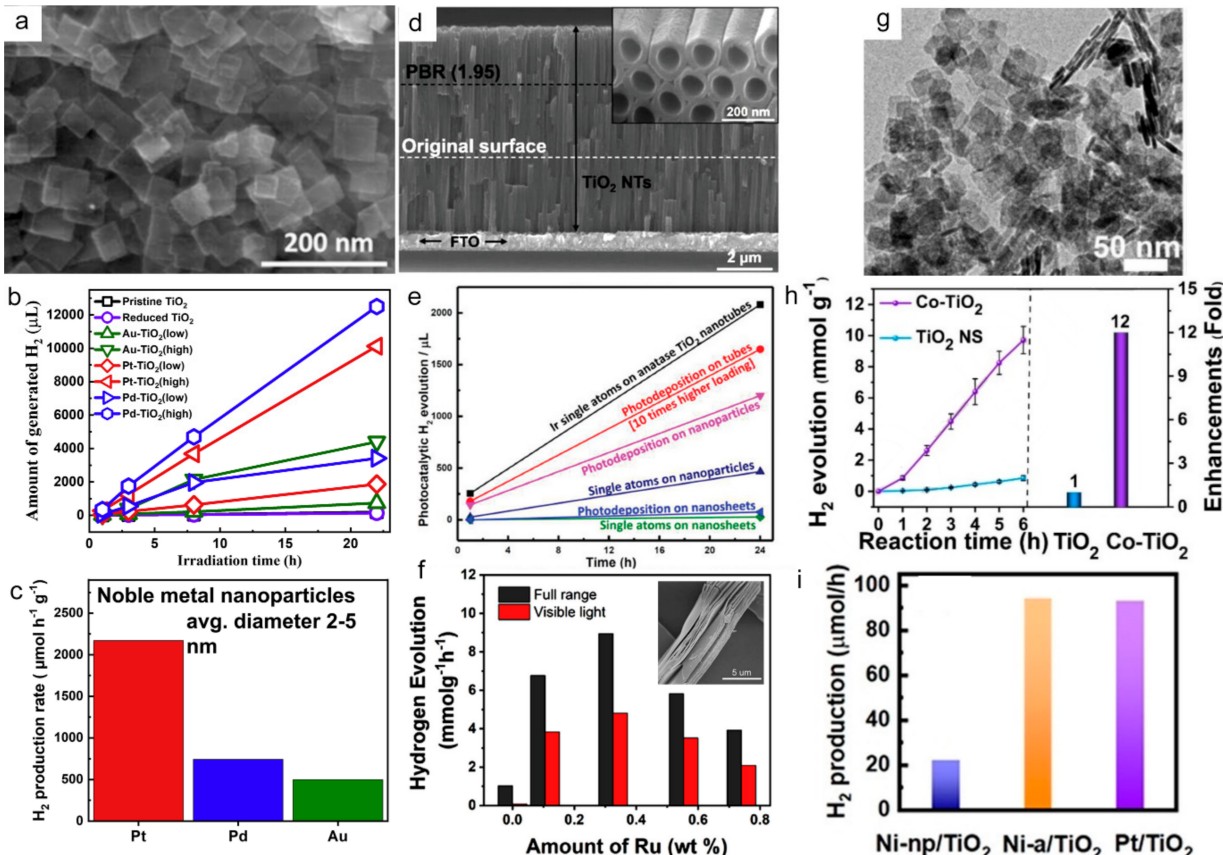

**Figure 12.** (**a**) SEM image of TiO$_2$ nanosheets. (**b**,**c**) Overview of the longtime H$_2$ evolution from TiO$_2$ nanosheets with different (low or high) noble metal SA loading (Au-TiO$_2$, Pt-TiO$_2$, and Pd-TiO$_2$) (**b**), H$_2$ evolution rate of TiO$_2$ nanosheets decorated with noble metal NPs with a diameter of ≈2–5 nm and loading of 1 wt.% (Pt, Pd, or Au, respectively) decorated NPs (**c**). Adopted with permission from Ref. [14] (**d**,**e**) SEM image of TiO$_2$ nanotubes (**d**), and photocatalytic H$_2$ evolution of Ir decorated TiO$_2$ samples. Adopted with permission from Ref. [134] (**e**) Time-dependent H$_2$ evolution and enhancement of evolution over Co-TiO$_2$ and TiO$_2$ NSs. Adopted with permission from Ref. [156] (**f**) H$_2$ production rate of Ru$_1$/TiNS with different amounts of Ru. Adopted with permission from Ref. [140]. (**g**,**h**) TEM images of Co-TiO$_2$ (**g**), and Time-dependent hydrogen evolution and enhancement of evolution over Co-TiO$_2$ and TiO$_2$ NSs. Adopted with permission from Ref. [156]. (**i**) Comparison of H$_2$ evolution activity on Ni-NP/TiO$_2$, Ni-SA/TiO$_2$, and Pt/TiO$_2$ with the same loading amount (~0.5 wt.%). Adopted with permission from Ref. [157].

Hwang et al. [158] decorated noble metal single-atoms of Pt, Pd, Rh, or Au on anatase TiO$_2$ NTs, using an immersion method leading to oxygen-coordinated M$^{\Delta+}$ state. A low SA loading of Pd on the TiO$_2$ NTs was much more effective as a co-catalyst for the photocatalytic H$_2$ generation in plain water than Pt, Rh, or Au SAs (or than corresponding noble metal NPs). The Pd-loaded catalyst provides an H$_2$ production rate of 0.381 μmol h$^{-1}$ cm$^{-1}$ at a density of 0.41 × 10$^5$ Pd atoms μm$^{-2}$. It was ascribed to suppression of H$_2$ and O$_2$ recombination reactions at individual SAs when used as co-catalysts in photocatalytic splitting of plain water.

Li et al. [140] prepared monolayered TiO$_2$ nanosheets, doped by Ru SAs (Ru$_1$/TiNS). Experimental characterizations and theoretical calculations reveal that the single-atom Ru$_1$ introduces an impurity energy level, allowing light absorption up to λ = 700 nm, and an O$_V$ around Ru$_1$ tends to be a charge trapping site, promoting rapid photogenerated electron separation and transportation. The Ru$_1$/TiNS demonstrated an 8-fold increase in H$_2$ production rate from 1.03 mmol h$^{-1}$ g$^{-1}$ for TiNS to 8.95 mmol h$^{-1}$ g$^{-1}$ under full radiation enabling visible-light H$_2$ production at 4.81 mmol h$^{-1}$ g$^{-1}$ rate. Zhang et al. [109]

prepared atomically dispersed Ru atoms decorated on multi-edged (ME) $TiO_2$ spheres. By the introduction of Ru SAs on the $TiO_2$ surface, several states in the bandgap were observed. The occupied states situated near the valence band results in reduced bandgap and easier excitation and transfer of photogenerated electrons to the Ru center, while the unoccupied states near the conduction band minimum accelerate the electron transfer for the activation of the water molecule. In comparison to Ru-$TiO_2$, ME-$TiO_2$@Ru showed reduced bandgap resulting in enhanced charge separation. In situ X-ray absorption spectroscopy was utilized to investigate the photoelectron migration. Upon light irradiation, the white line intensity reduces indicating that Ru species was quickly reduced after receiving an electron from $TiO_2$, i.e., photoelectron transport was enhanced. Furthermore, highly oriented crystals of ME-$TiO_2$@Ru with a sharp boundary accelerated charge transport to the Ru sites. As a result, the ME-$TiO_2$@Ru sample (Ru loading amount of 0.93 wt.%) showed the $H_2$ evolution rate of 323.2 $\mu$mol h$^{-1}$, which was 34-fold higher than that of pure ME-$TiO_2$.

Co, Cu, and Ni have been also investigated due to their high abundance and low cost. Wu et. al. [156] synthesized Co SAs on two-dimensional $TiO_2$ nanosheets (Co-$TiO_2$) for the photocatalytic $H_2$ evolution reaction. The atomic Co is incorporated in $TiO_2$ through Co-O bonding. The $\Delta G_{H*}$ for Co SA $TiO_2$ (2.104 eV) was lower than $TiO_2$ NSs (3.423 eV) indicating optimized binding H* and electron-proton acceptance. Theoretical calculations showed that the effective electronic coupling via Co-O coordination modulated the electronic structure, which prompted the electron transport, resulting in Co SA-$TiO_2$ (with 1.11 wt.% of Co loading) exhibiting an $H_2$ evolution rate of 1.682 mmol h$^{-1}$ g$^{-1}$, which was 12-fold higher than that of plain $TiO_2$ (Figure 12h).

Xiao et al. [157] synthesized atomic nickel co-catalyst on $TiO_2$ by a molten salt method. The liquid environment and space confinement effect of the molten salt leads to atomic dispersion of Ni ions on $TiO_2$, while the strong polarizing force provided by the molten salt promotes the formation of strong Ni-O bonds. Furthermore, Ni atoms were found to facilitate the formation of $O_V$ on $TiO_2$ during the process, which benefits the charge transfer and HER. The resultant photocatalyst exhibited ~4-fold $H_2$ generation enhancement as compared to the conventional Ni-NP/$TiO_2$ and comparable performance to that of the noble-metal Pt-NP/$TiO_2$ with the same loading amount (ca. 0.5 wt.%) (Figure 12i). Fan et al. [136] demonstrated that Ni/$TiO_2$ is visible light active due to the presence of atomically isolated Ni-oxo groups. The $(\equiv TiO)_2Ni(H_2O)_4$ species create small "*p*-type zones" strongly affecting the local electric structure of neighboring $TiO_6$ octahedra. The results indicated that Ni atomically isolated on $TiO_2$ is fully responsible for the visible light absorption and photocatalysis of Ni/$TiO_2$ composites. Under UV light irradiation, Ni acts as an electron-trapping and hydrogen-evolving site. Electrons photogenerated on $TiO_2$ were transferred to the Ni moieties via the T-O-Ni linkages formed at the NiO/$TiO_2$ interface. The visible light photocatalysis induced by heterobimetallic T-O-Ni linkages follows the physical mechanism of the metal-to-metal charge transfer, while hydrogen gas evolved at $O_V$ on the $TiO_2$ surface.

Lee et al. [159] synthesized Cu SA on $TiO_2$ (Cu/$TiO_2$) via the wrap-bake-peel process and investigated how the valence state change of isolated Cu atoms affects the overall photocatalytic reaction. The as-prepared Cu SA/$TiO_2$ is in a resting state, which is inactive (CT0 state). The CT0 state changes to a photoexcited state (CT1 state) by absorbing light, which generates electrons and holes. A photogenerated electron transfers from the conduction band of $TiO_2$ to the *d* orbital of the isolated Cu atoms, while the extra charge is compensated by oxygen protonation, resulting in a valence change of the redox-active isolated Cu atoms (CT2 state). The trapped electron at the Cu *d* orbital induces a polarization field, resulting in local $TiO_2$ lattice distortion around the isolated copper atoms (CT3 state). The resulting CT3 state has different photoelectrochemical properties enhancing photocatalytic $H_2$ generation. Importantly, the active CT3 state can be easily reverted to its original resting CT0 state when simply exposed to $O_2$ for a few minutes under dark conditions. To shed light on the role of valence state on photocatalytic effect, DFT studies were performed demonstrating the midgap states in $Cu_1$/$TiO_2$. Electron is localized on $d_{z2}$ and changes the valence state

of Cu atom accompanied by the addition of a proton to $TiO_2$ to balance charge. More importantly, the Cu single-atoms have an axial antibonding characteristic inducing a lattice distortion on $TiO_2$ by elongating the backside oxygen coordination tuning the initially sluggish $TiO_2$ to an active state. Furthermore, the wrap-bake-peel process was repeated with Co, Fe, Ni, Cu, and Rh single atoms to compare their overall photocatalytic performances, while the $Cu/TiO_2$ exhibited enhanced photocatalytic activity in the $H_2$ production rate of 16.6 mmol $g^{-1}$ $h^{-1}$ at optimal (0.75 wt.%) Cu loading, which was ~34-fold higher in comparison to plain $TiO_2$ counterparts. Zhang et al. [160] developed a re-encapsulation strategy to stabilize Cu SA on $TiO_2$ by incorporating Cu initially in metal-organic-framework (MOF) MIL-125 followed by its calcination to form Cu $SA/TiO_2$. The obtained MOF structure with a large surface area maximizes exposed sites for Cu SA immobilization on $TiO_2$ achieving a Cu SA loading of ~1.5 wt.%. The optimized Cu $SA/TiO_2$ sample shows the photocatalytic $H_2$ evolution rate of 101.7 mmol $g^{-1}$ $h^{-1}$ with an apparent quantum efficiency (*AQE*) of 56% at $\lambda$ = 365 nm irradiation.

*6.2. Photocatalytic $CO_2$ Reduction*

By mimicking photosynthesis, photocatalytic conversion of $CO_2$ into value-added products is a vital approach to addressing the problem of increasing $CO_2$ emissions as well as the energy crisis. However, the performance of current photocatalysts is still far from satisfactory, while enhanced stability, selectivity, and photocatalytic activity of SACs have attracted increasing attention for photocatalytic $CO_2$ reduction. On Pt, $CO_2$ reduction to CO is possible. However, further reduction to hydrocarbon such as $CH_4$, and $C_2H_6$ is difficult due to strong CO adsorption [161], while it is possible to reduce $CO_2$ to hydrocarbons with Pt SA by tuning adsorption of reactant molecules. Cai et al. [162] immobilize Pt SA via photoreduction method by utilizing oxygen vacancy defects on $TiO_2$ nanotubes obtained by annealing of titanic acid nanotubes. The number of oxygen vacancies was controlled by annealing at various temperatures ($O_V$-$TiO_2$). The formation of SAs was confirmed by HAADF-STEM and the EPR measurements that displayed an effect of temperature on the concentration of $O_V$, while $O_V$-$TiO_2$ annealed at 500 °C showed an $O_V$ maximum. The underlying mechanism was studied by DFT calculations on anatase $TiO_2$ (101) surface with and without $O_V$. For Pt SA adsorbs on a clean a-$TiO_2$ (101) surface, the $O_{2c}-O_{3c}-O_{2c}$ site is the most favorable one with an adsorption energy of $-2.89$ eV. It was revealed that the adsorption energy of Pt SA becomes more negative in the presence of surface and subsurface oxygen vacancies. Such surface vacancies give rise to more donor electron density on the $TiO_2$ surface, thereby facilitating charge transfer, and hence improving photocatalytic activity. The photocatalytic activity of the developed catalyst was assessed through HER and $CO_2$ photocatalytic reduction. $O_V$-$TiO_2$ showed increased photocatalytic activity due to suppressed charge carrier recombination, while the sample $O_V$-$TiO_2$-500 with the maximized $O_V$ concentration showed the highest $H_2$ production rate (1.077 mmol $h^{-1}$) with an apparent quantum efficiency (*AQE*) of 21.7% under $\lambda$ = 365 nm illumination. However, the *AQE* for photocatalytic reduction of $CO_2$ was 0.064% only. This is because $CO_2$ reduction to produce $CH_4$ is an eight-electron process, which requires gathering multiple photogenerated charge carriers with a long lifetime [163].

Lee et al. [164] demonstrated that the introduction of another Cu atom nearby led to inferior photocatalytic performance for $CO_2$ reduction. While Cu NP-based co-catalyst demonstrated the photocatalytic performance enhancement with the increase in the co-catalyst loading (typically 1–3 wt.%), which was attributed to an increase in the active sites, $Cu_1/TiO_2$ exhibited the highest $CO_2$ reduction performance at a low Cu concentration (0.21 wt.%), which was significantly decreased with Cu concentration increase. This led the authors to the conclusion that factors other than active sites are responsible for the high catalytic performance of $Cu_1/TiO_2$. According to theoretical studies, the Cu atom in $Cu_1/TiO_2$ has a well-localized electron, leading to strong Lewis basicity of the Cu atom. Moreover, the localization of the electrons renders Cu-$O_4$ complex seesaw ($C_{2v}$) structure. This charge localization to the active Cu center and distorted Cu-$O_4$ structure offers a

binding pocket for chemisorption of $CO_2$ molecules resulting in enhanced photocatalytic $CO_2$ reduction. In contrast, the d electron in $Cu_2/TiO_2$ is delocalized with a nearby Cu atom, leading to poor Lewis basicity of the Cu atom; while the $Cu-O_4$ complex after structural relaxation attains a more symmetrical square planar ($D_{4h}$) structure. As a result, $Cu_2/TiO_2$ is electronically and sterically unfavorable for $CO_2$ adsorption.

As in the case of $H_2$ generation, Cu-based co-catalysts have been found effective for photocatalytic $CO_2$ reduction with $H_2O$ being a low-cost alternative to other noble metals [165,166]. Jiang et al. [167] prepared atomically dispersed Cu on ultrathin $TiO_2$ nanosheets for photocatalytic reduction of $CO_2$ to CO in an aqueous solution. The Cu SA was prepared by a photoreduction method, i.e., under solar irradiation photoelectron from $TiO_2$ conduction band reduces $Cu^{2+}$ from copper nitrate precursor to SA $Cu^0$. The photogenerated electron is then transferred from $TiO_2$ to $Cu^0$ to reduce $CO_2$ to CO and $O^{2-}$, while $H_2O$ is oxidized to $O_2$ and $H^+$. The $Cu^0$ aids the separation of electron-hole charge carriers to enhance photocatalytic reduction of $CO_2$ and oxidizes to $Cu^{\Delta+}$ (when $\Delta$ is <2). The oxidized $Cu^{\Delta+}$ cannot be reduced by photoelectrons from $TiO_2$, as its reduction potential is higher than the conduction band energy of $TiO_2$ but can be reactivated by opening the reactor to air. Yuan et al. [168] decorated atomically dispersed Cu species over mesoporous $TiO_2$ ($mTiO_2$) nanobeads for photocatalytic $CO_2$ reduction in the gas phase with $H_2O$. The combination of in situ measurements, i.e., EPR, EXAFS, and XANES, was applied to study the reaction mechanisms for the photoreduction of $CO_2$ with $H_2O$ over the $Cu-mTiO_2$ catalyst. It was proposed that the gas mixture of $CO_2$ and $H_2O$ vapor be diffused into the porous structure of $mTiO_2$ and adsorbed on the catalyst surface. Upon light illumination, $mTiO_2$ is bandgap excited to generate electron-hole pairs, and the grafted Cu species and surface hydroxy groups trap electrons and holes, respectively, thus, boosting the spatial separation of charge carriers. The initial Cu(II) species can be gradually reduced to Cu(I) by the electrons and further to Cu(0), while Cu(0) can trap the holes to be re-oxidized to Cu(I) and further to Cu(II). Since Cu species are more efficient for attacking electrons, the reduction process was faster than the oxidation process, thus ultimately transferring the Cu species to Cu(0), which dynamically catalyzed the photoreduction of $CO_2$. Adsorbed $H_2O$ and surface hydroxyl groups on the surface of $Cu-mTiO_2$ were subsequently activated by photoinduced holes to form $O_2$ and protons. Finally, the photoelectrons were transferred to react with $CO_2$/CO and/or protons to form $CH_4$. Lee et al. [164] investigated the atomic-level photocatalytic process on the $TiO_2$ surface using uniformly stabilized Cu SAs highlighting that the redox-mediating property of the support can directly affect electronic interaction at the metal atom–support interface. In the case of $Cu_1/TiO_2$, the presence of a single Cu atom partially oxidizes an adjacent surface O atom to stabilize the metal–support interface, and thereby, spontaneously generates an oxygen vacancy near the Cu SA. $CO_2$ is chemically adsorbed on the $O_V$ site by strong charge transfer of the localized electron in the Cu SA, while an isolated Cu atom on the $TiO_2$ surface plays two important roles in the photocatalytic $CO_2$ reduction: (i) it promotes the $O_V$ formation and replenishment by enhancing the surface reducibility of the surrounding $TiO_2$, and (ii) it induces electron localization to the active Cu center that can offer a binding pocket for favorable chemisorption of a $CO_2$ molecule. However, the charge localization to the Cu atoms is disturbed when additional Cu atoms are closely located, which directly affects the $CO_2$ photoreduction performance. Such optimized $Cu_1/TiO_2$ photocatalysts exhibited a 66-fold enhancement in $CO_2$ photoreduction performance compared to the pristine $TiO_2$.

In addition, bimetallic co-catalysts, such as Pt-Au and Cu-Au, have been explored owing to the electronic structure of active sites, which could enhance photoactivity toward $CO_2$ reduction [169,170]. Yu et al. [171] demonstrated the synergistic effect of metal alloy NPs and SAs for efficient photocatalytic reduction of $CO_2$. They synthesized Cu SA and Au-Cu metal alloy NPs deposited on $TiO_2$ and tested photoactivity by $CO_2$ photoreduction. HAADF–STEM and TEM revealed the presence of Cu SAs as well Au-Cu alloy NPs. The Cu SAs + Au-Cu alloy NPs/$TiO_2$ catalyst showed catalytic performance with a formation rate of 3578.9 and 369.8 $\mu mol\ g^{-1}\ h^{-1}$ for $CH_4$ and $C_2H_4$, respectively, using Cu SAs $TiO_2$,

Au SAs $TiO_2$, and Au-Cu alloy $TiO_2$. DFT calculations reveal that after adsorption of Cu on $TiO_2$ electron is transferred from Au-Cu alloy NP to Cu SAs, therefore, Cu SAs acquired a negative charge. Furthermore, the free energy of adsorption Cu SA + Au-Cu alloy NPs were found to be more negative than after adsorption of Cu SA and Au-Cu alloy NPs suggesting spontaneous adsorption of $CO_2$ and $H_2O$ on these surfaces. The in situ FTIR spectra were conducted to identify the key intermediates and reaction pathways, while the Cu SA + Au-Cu alloy NPs, as well as Cu SA, showed almost identical pathways. Furthermore, free energy calculation showed that Cu SAs + Au-Cu alloy $NPs/TiO_2$ reduces the energy barrier for the formation of key intermediates $CO_2$*, COOH*, $CH_2OH$*, and $CH_2$*. Therefore, enhancing photoactivity in comparison to Cu $SA/TiO_2$.

Pan et al. [172] prepared binary Pt-Au SACs by anchoring Pt and Au on the $O_V$ of self-doped $TiO_2$ nanotubes support (Pt-Au/R-TNTs). The SAs Pt-Au/R-TNTs with a composition of 0.33 wt.% of SAs metals showed maximal performance in the photocatalytic $CO_2$ reduction, while the $CO_2$ molecule was initially protonated to form $\bullet CH_3$, and converted further into $CH_4$ and a C–C coupled product of $C_2H_6$ with production rates of 360.0 and 28.8 $\mu mol\ g^{-1}\ h^{-1}$ and apparent quantum yield (AQY) of 15.2 and 2.7%, respectively. The efficiency was about 5.5 and 149 times higher than that of the Pt-Au/TNTs and TNTs, respectively. The improved performance was attributed to the enhancement in the separation of photo-generated electron-hole pairs and charge-carrier transmission by the metal support interactions (MSI) of covalent bonding. Furthermore, the pathway of photocatalytic $H_2O$ oxidation changed from $\bullet OH$ generation to $O_2$ evolution upon loading of Pt and Au, resulting in inhibition of the self-oxidization of the photocatalyst.

### 6.3. Ammonia Production

Reactive or so-called "fixed" nitrogen, the form necessary for living organisms and used as basic biological building blocks is available in a relatively small amount as compared to $N_2$ in nature due to the high dissociation enthalpy of the triple bond of the $N_2$ molecule [173]. In fact, ammonia ($NH_3$) production is one of the largest-volume industrial chemicals synthesized in the world [174]. Recently, ammonia gained significant interest as a fueling vector being considered a candidate for power transport, producing energy, and supporting heating applications [175]. So far, ammonia is mostly relying on the Haber-Bosch (H-B) synthesis process [7], while its high energy consumption and carbon footprint are responsible for ~2% of the world's greenhouse gas emissions [176–178]. The photocatalytic process proposes a viable route for $NH_3$ synthesis. However, the activity of common photocatalysts is still low for industrial applications. In this sense, SACs present a feasible direction due to their high utilization efficiency and photocatalytic activity.

Ru is considered one of the best co-catalysts for nitrogen reduction reaction (NRR) [179]. Liu et al. [44] fabricated $TiO_2$ nanosheets decorated by Ru SAs enabling photocatalytic reduction of $N_2$ to $NH_3$ under xenon lamp illumination. DFT calculations indicated that Ru SAs were stabilized by $O_V$. XANES spectra of Ru SA on $TiO_2$ showed an adsorption threshold close to $RuCl_3$, indicating that Ru has a valence state close to 3+. In addition, the Ru K-edge EXAFS spectrum had a predominant peak at ca. 1.5 Å, while no peak was observed at ca. 2.3 Å confirming the atomic nature of Ru rather than Ru cluster or NP, while in the case of pristine $TiO_2$ without oxygen vacancies only one peak ca. 2.3 Å was observed indicating aggregation of Ru into NPs. The optimized composite catalyst containing 1 wt.% of Ru showed an improved ammonia generation rate of up to 56.3 $\mu g\ h^{-1}\ g^{-1}$, which was 2-fold higher than that of pure $TiO_2$ nanosheets. Isolated Ru atoms, located at the oxygen vacancies of $TiO_2$, helped weaken the hydrogen evolution promoting chemisorption of $N_2$, and improving charge carrier separation, leading to enhanced $N_2$ photoreduction (Figure 13a–c). Niu et al. developed a Ru-based SAC anchored on defect-rich $TiO_2$ nanotubes (Ru SA Def-$TiO_2$) (Figure 13d). The Ru SA Def-$TiO_2$ greatly promoted electron transfer due to the formation of the ligand-to-metal charge-transfer state (LMCT), which was found to be responsible for its catalytic activity, promoting the transfer of photoelectrons from Def-TNs to the Ru-SAs center and be subsequently captured by

Ru SAs (Figure 13e). The photocatalytic $N_2$ fixation tests were conducted in $N_2$-saturated water with full-spectrum irradiation under ambient conditions, without using any sacrificial agents. Ru-SAs Def-TNs achieved a catalytic efficiency of 125.2 µmol h$^{-1}$ g$^{-1}$, which was ~6 and 13-fold higher than Ru NPs and Def-TNs, respectively. The Ru-SAs/Def-TNs catalysts exhibited stability after five photocatalytic cycles (Figure 13f).

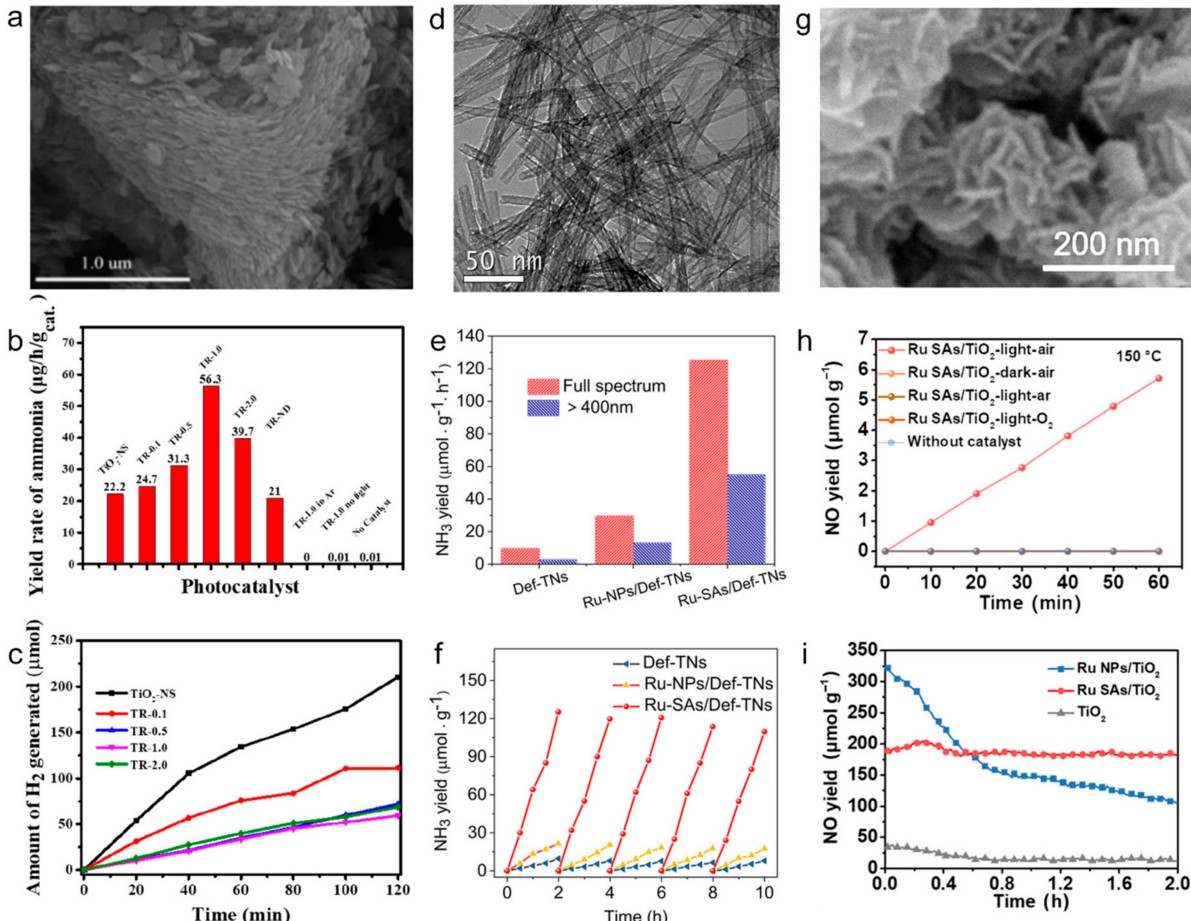

**Figure 13.** (**a**) SEM image of $TiO_2$ nanosheets. (**b**) The yield rates of ammonia and (**c**) the amounts of $H_2$ generated over $TiO_2$ nanosheets with various amounts of Ru. Adapted with permission from Ref. [44]. (**d**) HRTEM images of Def-TNs, (**e**) Comparison of the total $NH_3$ yields in the first 2 h under full-spectrum or visible-light ($\lambda > 400$ nm) irradiation, and (**f**) Quantitative determination of the $NH_3$ yields as a function of irradiation time under full-spectrum irradiation (300 W Xe lamp) and five cycling tests for the three samples, i.e., Def-NTs, Ru NP/Def-NTs and Ru SA/Def-NTs. Adapted with permission from Ref. [42]. (**g**) SEM image of Ru SAs/$TiO_2$. (**a**) TEM (**h**) Performances and verification experiments of NO production via the photo-oxidation of $N_2$ over Ru SAs/$TiO_2$. (**h**) Quantitative determination of generated NO over Ru SAs/$TiO_2$ under different conditions at 150 °C. (**i**) Time-dependent NO yield over different samples at 250 °C. Adapted with permission from Ref. [180].

Thermally assisted photo-driven nitrogen oxidation to nitric oxide (NO) using air as a reactant is a promising route to displace the traditional NO synthesis industry accompanied by huge energy expenditure and greenhouse gas emissions. Huang et al. [180] employed atomically dispersed Ru-decorated $TiO_2$ nanosheets (Ru SAs-$TiO_2$) for thermally assisted photocatalytic oxidation of $N_2$ into NO (Figure 13g). The photocatalytic oxidation of the $N_2$ reaction was examined at a temperature ranging from 150 to 300 °C, while 250 °C was optimal owing to long-time stability (Figure 13h,i). The Ru SAs-$TiO_2$ exhibited performance toward NO formation with a 192 µg h$^{-1}$ g$_{cat}$$^{-1}$ and attained a quantum efficiency of 0.77%

at $\lambda$ = 365 nm. Theoretical simulations revealed that the formation of $(N-O)_2$-Ru is the rate-determining step for the photo-oxidation of $N_2$, which was further confirmed by the XPS and DRIFTS measurements. Furthermore, the activity of Ru SAs-$TiO_2$ remained constant over the time of the experiment, while the activity of Ru NPs-$TiO_2$ decreased due to conversion into bigger $RuO_2$ NP, leading to the deactivation of the catalyst.

*6.4. Other Applications*

6.4.1. Photoelectrochemical Water Splitting

To improve the low photonic efficiency of the semiconducting materials owing mainly to the recombination of the excited charge carriers, the coupling of electrochemical and photocatalytic processes, i.e., photoelectrocatalysis, is commonly utilized [181]. Photoelectrocatalysis is based on the separation of photo-generated charge carriers by gradient potential, while the semiconductor is immobilized on a conductive substrate used as photoelectrode in a photoelectrochemical cell. The mechanism on the photoelectrode is similar to photocatalysis, i.e., photon absorption and charge separation, while the application of a bias potential is a way to control the Fermi level of a semiconductor, i.e., to affect the charge separation on the space charge layer (SCL) facilitating the reactions of charge carriers on the semiconductor surface. Considering an *n*-type semiconductor, the electric field produced by external bias causes electrons to migrate towards the conductive support, i.e., a counter electrode, while holes move towards the solution interface, thereby increasing the lifetime of the $e^-$-$h^+$ pairs, which leads to improved reaction rates [182].

Photoelectrochemical (PEC) water splitting has attracted great attention during the last decades due to the feasibility to be a cost-effective process in $H_2$ or solar fuel production doing it in a single step powered by solar energy. Despite significant efforts, low photoconversion efficiency, high charge transfer resistance, and fast recombination rate are the bottlenecks of semiconductor nanomaterials in the PEC water splitting process [183]. The achieved quantum efficiencies of PEC water-splitting cells are still insufficient for practical applications due to a lack of suitable candidate semiconductor-based materials but also because of long-term stability limitations, i.e., photocorrosion occurs under harsh photoinduced redox reaction conditions. Among available benchmark semiconductors, anatase $TiO_2$ is a vital candidate owing to its superior chemical stability and abundance, yet it suffers from the same limitation discussed above, i.e., intrinsic weak light-harvesting and low photocatalytic efficiency. An application of co-catalysts mainly in the form of NPs was widely studied to enhance the overall quantum efficiency of the $TiO_2$ under PEC conditions [184], while only a few studies have been published so far using a SAC approach for PEC water splitting.

Wang et al. [144] studied the (Rh + F) surface co-doping effect on anatase $TiO_2$ (101) for solar water splitting by DFT calculations. The CBM of the pure anatase $TiO_2$ (101) surface is higher by about 0.2 eV than the reduction potential of $H^+$/$H_2$, while its VBM is more positive by about 1.31 eV than the oxidation potential of $O_2$/$H_2O$. The electrochemical characteristics indicate a strong reduction/oxidation ability of pure $TiO_2$ but ineffectiveness in solar light harvesting. While $TiO_2$ is co-doped by (Rh + F), the VBM and CBM positions are just slightly perturbed suggesting that the strong oxidizing/reducing abilities of the co-doped $TiO_2$ surface remain, while the occupied and delocalized intermediate states, which prevent the recombination of photogenerated charge carriers, with a bandwidth of 0.35 eV appear in the bandgap reduce it to 2.14 eV. Cheng et al. [185] investigated the influence of Cu SAs on the light absorption and photocatalytic characteristics of anatase $TiO_2$ used for solar water splitting. By performing molecular dynamics simulations, the authors investigated non-radiative charge trapping and recombination in Cu-SA-doped anatase $TiO_2$ acting as a photocatalyst. The atomistic analysis showed that the activity arises from the $H_{ads}$-Cu/$TiO_2$ species rather than Cu/$TiO_2$. Without the co-adsorbed H, the Cu dopant creates two trap states inside the $TiO_2$ bandgap facilitating rapid non-radiative losses. A shallow trap near the VB maximum arises from O 2*p* orbitals next to the Cu dopant but has no contributions from the dopant itself. A deep midgap trap is supported

by the *d*-orbital of the Cu dopant and neighboring O atoms in the *ab* plane. Photoinduced electron transfer and protonation of the Cu dopant site on the $TiO_2$ surface push the shallow trap state into the VB, extending the charge carrier lifetime by an order of magnitude. The simulations showed that the local structure undergoes a noticeable distortion upon Cu doping, while thermal fluctuations lead to a dynamic balance of adsorption and desorption of proton around the active site on the $Cu/TiO_2$ surface and drive water splitting to proceed persistently. Although H adsorption eliminates the shallow trap, it introduces faster motions and enhances fluctuations of the midgap state energy, making the midgap trap a stronger non-radiative relaxation center.

So far, the influence of SACs on PEC water splitting was investigated mainly theoretically to predict the favorable effect of SA co-catalyst on PEC characteristics of $TiO_2$ nanostructures. Pang et al. [186] demonstrated the role of atomically dispersed bismuth (Bi) assembled on $TiO_2$ nanorods for PEC water splitting. The binding energy peaks of $Bi/TiO_2$ show a slight positive shift as compared to those of pristine $TiO_2$ precursors. The binding energy of Bi in $Bi/TiO_2$ was smaller than that of Bi(III) in $Bi_2O_3$, but was larger than that of Bi(0), demonstrating the existence of Bi atoms in low-valent states ($Bi^{\Delta+}$, $0 < \Delta < 3$) and indicating that Bi can form the Bi-O-Ti bond with the $TiO_2$ support facilitating charge transfer among Bi, Ti, and O elements [187]. The photocurrent density of the atomically dispersed $Bi/TiO_2$ with the optimal Bi amount was estimated as 1.65 mA cm$^{-2}$ at the applied potential of 1.23 $V_{RHE}$ being remarkably higher than that of pristine $TiO_2$ (0.42 mA cm$^{-2}$ at 1.23 $V_{RHE}$). However, the detailed mechanism of PEC enhancement lacks in this study, and some explanations are misleading. Therefore, more experimental evidence is required to further understand the beneficial role of Bi SAs in the PEC water splitting process.

6.4.2. Synthesis and Degradation of Organic Compounds

Products of organic synthesis are used as intermediates in various branches of the chemical industry, while the technological limitations resulting from physicochemical factors as well as economic and safety requirements, push forward synthesis methods to simplify the reaction pathways to minimize overall process expenses and environmental pollution. In this sense, photocatalysis offers a potential solution since it is environmental-friendly and economically favorable, while its overall quantum efficiency should be enhanced. Below, we demonstrate the effectiveness of SA photocatalysts that has been explored so far to synthesize several organic chemicals.

Zhou et al. [46] explored a series of noble metals (Ru, Rh, Pd, Ag, Os, Ir, Pt, and Au) SA-loaded $TiO_2$ for photocatalytic reformation of acetone into 2,5-hexanedione (HDN) (Figure 14a). The photogenerated holes activate the C-H bond in the methyl group of acetone to form $CH_3COCH_2{}^{\bullet}$ radical, the two $CH_3COCH_2{}^{\bullet}$ radicals combine to form HDN, while photogenerated electrons reduce two H species forming $H_2$. The IR peaks at 2921 and 2852 cm$^{-1}$ attributed to the C-H bond in the methyl group of acetone were found on Pt SA-$TiO_2$ suggesting the formation of intermediate $CH_3COCH_2{}^{\bullet}$ radicals on these surfaces. The DFT calculated results suggested that Pt SA-$TiO_2$ has the lowest reaction barrier for acetone dehydrogenation reaction (Figure 14b), while the time-resolved photoluminescence (TR-PL) spectroscopy showed a prolonged lifetime in the case of Pt SA-$TiO_2$ (1.09 ns) in comparison to Pt NP-$TiO_2$ (0.73 ns). The enhanced charge separation of Pt SA was ascribed to enhance the electron capture ability of unoccupied 5*d* states. ATR-IR spectrum was performed to find out the key intermediate and reaction pathway. In situ electron spin resonance (ESR) measurements confirm $CH_3COCH_2{}^{\bullet}$ on Pt SA-$TiO_2$ acting as a key intermediate for the production of HDN by C-C coupling. Indeed, among examined metals, Pt SA-$TiO_2$ exhibited the highest photocatalytic activity of 3.87 mmol g$^{-1}$ h$^{-1}$ with a selectivity of 93% (Figure 14c), while the photocatalytic stability was confirmed by the HDN-production activity of Pt SA-$TiO_2$ up to 16 h in four circles.

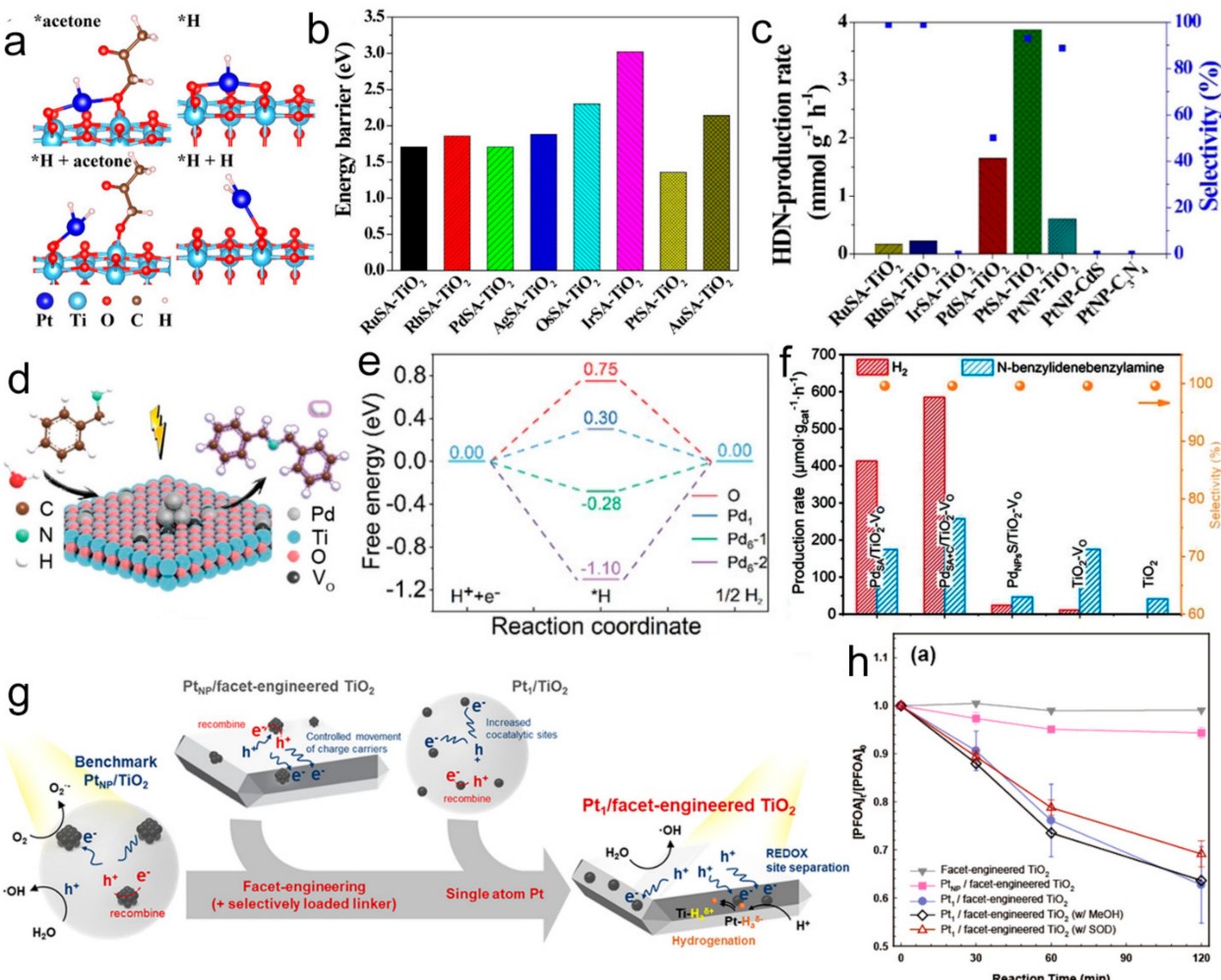

**Figure 14.** (**a**) Reaction path of acetone dehydrogenation, and (**b**) the corresponding energy barrier on MSA-TiO$_2$ (M = Ru, Rh, Pd, Ag, Os, Ir, Pt, and Au). (**c**) Photocatalytic HDN-production activity of as-prepared samples from acetone. Adapted with permission from Ref. [46]. (**d**) Schematic diagram of the photocatalytic reaction, (**e**) Free energy profiles for H adsorption on different sites of Pd$_1$+Pd$_6$/TiO$_2$-O$_V$ model: O-site near O$_V$, Pd$_1$ site, Pd$_6$-1 site at the Pd$_6$-TiO$_2$ interface, Pd$_6$-2 site on the top of Pd$_6$ cluster, (**f**) production rates of H$_2$ (red) and N-benzylidenebenzylamine (blue) over different photocatalysts. Adapted with permission from Ref. [188]. (**g**) Photocatalysis mechanisms of Pt NP-loaded TiO$_2$, Pt SA-loaded TiO$_2$, Pt NP-loaded facet-engineered TiO$_2$, and Pt SA-loaded facet-engineered TiO$_2$. (**h**) Photocatalytic degradation of perfluorooctanoic acid (PFOA) using Pt$_1$/facet-engineered TiO$_2$. Adapted with permission from Ref. [189].

Wang et al. [188] reported that the coexistence of Pd SA, cluster, and oxygen vacancy has favorable activity toward hydrogen evolution and selective oxidation of benzylamine (Figure 14d). The Pd SA/TiO$_2$-O$_V$ was synthesized via pyrolysis of the Pd$^{2+}$/MIL-125-NH$_2$ precursor. The concentration of Pd SAs, clusters, and O$_V$ was optimized by annealing Pd SA/TiO$_2$-O$_V$ in H$_2$/Ar atmosphere at 200 °C. To evaluate the phenomenon, DFT simulation was performed using Pd SA+C/TiO$_2$-O$_V$. It was revealed that the *d*-band center is downshifted and had a higher electron affinity than the electron transfer from TiO$_2$ to the Pd cluster. Furthermore, the active site in Pd$_1$ + Pd$_6$/TiO$_2$-O$_V$ was evaluated from $\Delta G_{H*}$ calculation. The Pd$_6$-1 site neighboring TiO$_2$ in Pd$_1$ + Pd$_6$/TiO$_2$-O$_V$ showed an optimal $\Delta G_{H*}$ value (−0.28 eV) (Figure 14e). The interaction between the Pd cluster and TiO$_2$ results in depletion of *d*-orbital electron density at the Pd$_6$-1 site, which is intermediate compared to the Pd$_6$-2 site and Pd$_1$ site. Meanwhile, the interaction of benzylamine at the O$_V$ site facilitates

the activation and selective oxidation of benzylamine on the hole ($h^+$)-accumulated $TiO_2$. The interaction of water molecules and benzylamine was studied by DFT calculations on $Pd_1/TiO_2$-$O_V$ and $Pd_1 + Pd_6/TiO_2$-$O_V$, revealing that it is greater in $Pd_1 + Pd_6/TiO_2$-$O_V$ as compared to $Pd_1/TiO_2$-$O_V$, resulting in the higher selectivity of $Pd_1 + Pd_6/TiO_2$-$O_V$ toward benzylamine oxidation as compared to $H_2O$. The optimized Pd SA + C/$TiO_2$-$O_V$ sample exhibited the production rate of $H_2$ (585.4 μmol $g^{-1}$ $h^{-1}$) and benzylidenebenzylamine (257.3 μmol·$g^{-1}$·$h^{-1}$) with the selectivity toward N-benzylidenebenzylamine formation of 99.9% compared to other samples (Figure 14f).

Zhang et al. [190] reported Fe SA supported $TiO_2$ hollow microspheres catalyst ($TiO_2$ HMs) for photocatalytic oxidation of NO for pollutant degradation. The adsorption edge of $Fe_1/TiO_2$-HMs between FeO and $Fe_2O_3$ suggests Fe has an oxidation state of +2 and +3, while Fourier transformed k3-weighted EXAFS demonstrated a peak at ≈2.5 Å, which corresponds to Fe-Ti bond suggesting Fe is dispersed in the form of a SA. In addition, the hybridization of Fe, Ti, and O forms shallow acceptor levels as evident from the visible light absorption of the $Fe_1/TiO_2$-HM samples. Based on differential charge analysis, electrons are transferred from Fe to the bonded Ti atoms facilitating the adsorption to activate NO and $O_2$ at Fe and bonded Ti sites, respectively. After the incorporation of Fe on $TiO_2$, the adsorption energy increased from −0.32 to −4.13 eV, O-O bond length increases from 1.21 to 1.40 Å and Bader charge increased from −0.23 to −0.73 e, suggesting that the incorporation of Fe SAs benefits the adsorption and activation of $O_2$ on the surface of $TiO_2$-HMs and of NO on $Fe_1/TiO_2$. As a result, the $Fe_1/TiO_2$-HMs sample exhibited a higher NO removal rate of ~48%, while the pristine $TiO_2$-HM sample showed a rate of ~21%. According to the IR measurements of the adsorbed nitrogen oxides species, the path of NO oxidation over $TiO_2$-HMs was as follows: $NO \rightarrow NO_2^- \rightarrow NO_2 \rightarrow NO_3^-$, while the introduction of atomically dispersed Fe results in another NO oxidation pathway: $NO \rightarrow N_2O_2^- \rightarrow NO_2^- \rightarrow NO_2 \rightarrow NO_3^-$. Here, $Fe_1/TiO_2$-HMs suppress the formation of unwanted $NO_2$ compared to $TiO_2$-HMs (the selectivity of $NO_2$ was 1.40% and 4.43% for $Fe_1/TiO_2$-HMs and $TiO_2$-HMs, respectively).

Weon et al. [189] investigated a controlled loading of Pt SA on the reduction sites of $TiO_2$. Facet-engineered $TiO_2$ that combines reductive {101} facets and oxidative {001} facets was used to efficiently separate the oxidation and reduction centers of $TiO_2$ [191]. Since the band position of the {001} facet on anatase $TiO_2$ is higher than that of the {101} facet, photogenerated electrons, and holes preferentially migrate to {101} and {001} facets, respectively (Figure 14g) [60,192]. Positively charged Pt SAs were selectively loaded onto the reductive sites of $TiO_2$ {101} facets to attract the photoinduced electrons efficiently and to augment both the reduction and oxidation efficiencies of the photocatalytic system. The latter enhances the number of holes, and consequently hydroxyl radicals, remaining on the sites of facet-engineered $TiO_2$, which was confirmed by the enhanced degradation of sulfamethoxazole and 2,4-dichlorophenoxyacetic acid. Furthermore, site-specifically-loaded Pt SAs produce surface hydrogen atoms enhancing hydrogen spillover onto the $TiO_2$ surface to cleave the C-F bond with the Ti-H bond achieving hydrodefluorination of perfluorooctanoic acid (PFOA) known as carcinogen and toxicant for humans being resistant to degradation by natural processes such as metabolism, hydrolysis, photolysis, or biodegradation (Figure 14h) [193]. An alternative pathway achieved on Pt SA-$TiO_2$ photocatalyst to harness the conduction band electrons for the degradation of PFOA is vital since most conventional advanced oxidation processes have been proven inefficient.

Trofimovaite et al. [194] investigated the promotion effect of single Cu(I) atoms Meso-$TiO_2$ materials for the application of methyl orange (MO) degradation and hydrogen production. Cu K-edge XAS was performed to find out the local environment and coordination number of Cu species. Least-squares spectral fitting showed the presence of Cu(II) species for loadings ≥ 0.81 wt.%, a fit to CuO, while < 0.3 wt.% showed a fit to mononuclear Cu(I). Pre-edge feature characteristics were found for Cu loading ≥ 0.3 wt.%, suggesting that Cu(I) is present in ultra-low loading of Cu. EXAFS spectra of 0.1 wt.% revealed a peak around 1.94 Å corresponding to Cu-O scattering, while no Cu-Cu peaks were observed

suggesting that Cu(I) is anchored on the surface. Mass normalized initial rates of dye degradation increase with isolated Cu(I) species, while decreasing with bulk CuO. This was attributed to the creation of $O_V$ upon introducing Cu(I) species into the $TiO_2$ framework, which serves as an electron capturing site, thereby reducing carrier recombination and promoting degradation of dye via direct or indirect hydroxyl oxidation. A 6-fold improvement in the MO degradation rate was observed over 0.02 wt.% Cu/Meso-$TiO_2$ sample.

### 7. Summary and Outlook

To summarize this review, it is clear that the saga of single-atom co-catalysts is just in its infancy and will take a certain time until SACs will conquer the world of photocatalysis—hopefully, it won't take too long. So far, several $TiO_2$ SAC systems have been reported to achieve higher photocatalytic activity than their nanoparticle and small cluster counterparts. However, a basic understanding of the intrinsic role of SACs in the enhancement of photocatalytic performance and controllable synthetic strategies is required to push forward and finally implement this beneficial technology in real life. Furthermore, optimized loading of SACs for photocatalytic applications is still challenging. From one side, increased loading produces more active photocatalytic sites but from the other leads to agglomeration of deposited metal atoms and their deactivation. Furthermore, single-atom photocatalysts consist mostly of rare and expensive metals. Therefore, it is important to optimize their function in the photocatalytic process. Therefore, the future direction will be toward a novel strategy for the controllable synthesis of single-atom photocatalysts to achieve higher photocatalytic activity and reaction specificity by utilization of the theoretical mechanism studies by computational methods to rationally design electronic and optical characteristics of SA photocatalysts, the metal SA loading sites, and optimal loading content on the photocatalyst. In this context, computational methods aim to provide a synergetic understanding of the role of the hosting material in anchoring SAs, i.e., modification of electronic structure due to the local electronic environment providing a better understanding of overall photocatalytic performance. Yet, ultrafast spectroscopies, electrochemical, and/or electron microscopy techniques should be combined to study the energy transfer and trapping processes to link such computational methods to experimental observations. Moreover, as we discussed earlier, not all SAs equally contribute to the photocatalytic reactions, while others detach and agglomerate. Therefore, in situ techniques to observe structural evolution of SA sites during the photocatalytic process need to be developed to guide the rational design of photocatalysts. Finally, while a certain number of papers on SACs made of a single component have been reported, there is little literature on di-nuclear active sites as well as on bi- and multi-metallic single atom sites. Undoubtedly, such active sites may provide a synergetic local electronic environment, further enhancing the catalytic performance of developed photocatalytic systems.

**Author Contributions:** Conceptualization, U.K., A.B.T. and P.S.; writing—original draft preparation, U.K. and A.B.T.; writing—review and editing, U.K., A.B.T. and P.S.; supervision, P.S. All authors have read and agreed to the published version of the manuscript.

**Funding:** This research was funded by Operational Program Research, Development and Education (European Regional Development Fund), and the Erlangen DFG cluster of excellence, Engineering of Advanced Materials (EAM) are greatly acknowledged for financial support. P.S. and A.B.T would like to acknowledge the DFG (grant number 442826449; SCHM 1597/38-1 and FA 336/13-1) for financial support. The authors would like to acknowledge the Operational Program Research, Development and Education (European Regional Development Fund, Project No. CZ.02.1.01/0.0/0.0/15_003/0000416 of the Ministry of Education, Youth and Sports of the Czech Republic) for financial support.

**Conflicts of Interest:** The authors declare no conflict of interest.

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
