# Peer review of "Single-Atom Co-Catalysts Employed in Titanium Dioxide Photocatalysis"

_catalysts, doi:10.3390/catal12101223_

Round 1

Reviewer 1 Report

This manuscript wants to present a review of single-atom co-catalysts employed in titanium dioxide photocatalysis. It briefly described the principal features of SAs, and gives an overview of most important examples of single-atom co-catalysts. Then, examples of single-atom co-catalysts used on TiO2 and their applications in photocatalysis were discussed and an outlook was provided for further exploring TiO2-based single-atom photocatalytic systems. There are still some issues should be addressed before it is accepted by Catalysts.

1.      The methods of synthesis, the first key step, of the research of single-atom co-catalysts used on TiO2 should be well classified and it is necessary to post the diagrams for better understanding. Here is one of the unique methods: Angewandte Chemie International Edition, 2021, 60(29): 16009-16018; Journal of catalysis, 2014, 320: 147-159; Journal of catalysis, 2012, 289: 88-99.

2.      Novel catalytic properties shown in single-atom catalysts were usually ascribed to the unique local electronic structure different from the clusters or nanoparticles, and more words are needed to summarize the uniqueness of the electronic structure.

3.      The figures of some examples posted in the manuscript couldn’t show the explicit structure of single-atom co-catalysts and were in need of better selection.

4.      The structure of the manuscript still needs to be optimized.

Author Response

Reviewer 1

We thank the reviewer for the positive feedback on our review. Below please find attached the detailed answers to the reviewer’s comments.

     Comment: The methods of synthesis, the first key step, of the research of single-atom co-catalysts used on TiO2 should be well classified and it is necessary to post the diagrams for better understanding. Here is one of the unique methods: Angewandte Chemie International Edition, 2021, 60(29): 16009-16018; Journal of catalysis, 2014, 320: 147-159; Journal of catalysis, 2012, 289: 88-99.

     Answer. We thank the reviewer for raising this comment. The methods forming SAs are described in the review in view of TiO2 photocatalyst and different metals as SACs. We completely agree with the reviewer that there are various approaches to forming SAs on a variety of semiconducting materials. In the revised review, we refer the reader to the extended reviews that were published in recent years with an emphasis on the synthesis of single atoms. We also added the suggested references to the synthesis section on Pages 10 and 13-14.

     Comment: Novel catalytic properties shown in single-atom catalysts were usually ascribed to the unique local electronic structure different from the clusters or nanoparticles, and more words are needed to summarize the uniqueness of the electronic structure.

     Answer. We totally agree with the reviewer that there is a substantial difference in local electronic structure between NP, clusters, and SAs that influences the photocatalytic pathways, it is discussed in Section 5.2.

     Comment: The figures of some examples posted in the manuscript couldn’t show the explicit structure of single-atom co-catalysts and were in need of better selection.

     Answer. We thank the reviewer for pointing this out. In the revised manuscript, we updated and revised the figures. As the reviewer knows, it is quite difficult to obtain a good resolution of single-atom even by state-of-the-art TEM imaging. In the SEM images, it is impossible to see SA decoration at all. But in the text, SEM images provide information on the photocatalyst structure rather than on SA decoration.

     Comment: The structure of the manuscript still needs to be optimized.

     Answer. The review is structured to emphasize the SA phenomenon in photocatalysis. We start the review from photocatalysis fundamentals with the advantages and challenges, followed by a co-catalyst approach to accelerate charge transfer and improve reaction selectivity by the introduction of SA active sites. This follows by synthesis and aspects that influence SAC activity. Finally, we present SA-based photocatalysts that were published during the last decade for various applications. In the revised manuscript we adopted this structure.

Reviewer 2 Report

Patrik Schmuki et al. gave a comprehensive review on the single atom co-catalysts employed in Titanium dioxide photocatalysts. It is meaningful and well written, could be accepted after minor revision.

1. The font size of the words in the maintext was not uniform, the authors should unify them.

2. Does SA play the roles in cocatalyst in all the mentioned catalysts? If no, it is better to revise the title of the review.

3. Section 3 was divided into three parts, “Co-catalyst approach”, “Schottky junction and its absence in SA co-catalysts”, “SAs and reaction selectivity”. The three parts were not parallel, it is better to reclassify them.

4. The authors should pay more attention to the summary and outlook section. What progress had been made in the area? What challenges are still remaining? What should we do to overcome the challenges?

Author Response

     We thank the reviewer for the positive feedback on our review. Below please find attached the detailed answers to the reviewer’s comments.

     Comment: The font size of the words in the main text was not uniform, the authors should unify them.

     Answer. We thank the reviewer for pointing this issue out. This happened due to the reformatting of the text by the journal. In the revised manuscript we corrected this issue.

     Comment: Does SA play the role of co-catalyst in all the mentioned catalysts? If not, it is better to revise the title of the review.

     Answer. The review focuses solely on SA co-catalysts developed during the last years on TiO2 for photocatalytic applications.

     Comment: Section 3 was divided into three parts, “Co-catalyst approach”, “Schottky junction and its absence in SA co-catalysts”, “SAs and reaction selectivity”. The three parts were not parallel, it is better to reclassify them.

     Answer. We thank the reviewer for this comment. We changed the section title to “Accelerating charge transfer and reaction selectivity of photocatalyst”.

     Comment: The authors should pay more attention to the summary and outlook section. What progress had been made in the area? What challenges are still remaining? What should we do to overcome the challenges?

     Answer. We thank the reviewer for this comment. In the revised manuscript, we updated the summary and outlook section.